# Sex-Specific Adaptations in Alzheimer’s Disease and Ischemic Stroke: A Longitudinal Study in Male and Female APP_swe_/PS1_dE9_ Mice

**DOI:** 10.3390/life15030333

**Published:** 2025-02-21

**Authors:** Klara J. Lohkamp, Nienke Timmer, Gemma Solé Guardia, Justin Shenk, Vivienne Verweij, Bram Geenen, Pieter J. Dederen, Lieke Bakker, Cansu Egitimci, Rengin Yoldas, Minou Verhaeg, Josine Kothuis, Desirée Nieuwenhuis, Maximilian Wiesmann, Amanda J. Kiliaan

**Affiliations:** 1Department of Medical Imaging, Anatomy, Radboud University Medical Center, Donders Institute for Brain, Cognition & Behavior, Center for Medical Neuroscience, Preclinical Imaging Center PRIME, Radboud Alzheimer Center, Nijmegen, The Netherlands; klara.lohkamp@radboudumc.nl (K.J.L.); n_m_timmer@hotmail.com (N.T.); gemma.soleguardia@radboudumc.nl (G.S.G.); shenk.justin@gmail.com (J.S.); vivienne.verweij@radboudumc.nl (V.V.); bram.geenen@radboudumc.nl (B.G.); jos.dederen@radboudumc.nl (P.J.D.); lieke.bakker@maastrichtuniversity.nl (L.B.); rcansuegitimci@gmail.com (C.E.); rengin.yoldas@gmail.com (R.Y.); minou_verhaeg@hotmail.com (M.V.); josinekothuis@mac.com (J.K.); maximilian.wiesmann@radboudumc.nl (M.W.); 2Alzheimer Center Limburg, Department of Psychiatry and Neuropsychology, Mental Health and Neuroscience Research Institute (MHeNs), European Graduate School of Neuroscience (EURON), Faculty of Health, Medicine and Life Sciences (FHML), Maastricht University, 6229 ER Maastricht, The Netherlands

**Keywords:** Alzheimer’s disease, stroke, sex differences, MRI, cerebral blood flow, cognition neuroinflammation, neuropathology

## Abstract

The long-term impact of stroke on Alzheimer’s disease (AD) progression, particularly regarding sex-specific differences, remains unknown. Using a longitudinal study design, we investigated transient middle cerebral artery occlusion in 3.5-month-old APP_swe_/PS1_dE9_ (APP/PS1) and wild-type mice. In vivo, we assessed behavior, cerebral blood flow (CBF), and structural integrity by neuroimaging, as well as post-mortem myelin integrity (polarized light imaging, PLI), neuroinflammation, and amyloid beta (Aβ) deposition. APP/PS1 mice exhibited cognitive decline, white matter degeneration (reduced fractional anisotropy (FA) via diffusion tensor imaging (DTI)), and decreased myelin density via PLI. Despite early hypertension, APP/PS1 mice showed only sporadic hypoperfusion. Cortical thickening and hippocampal hypertrophy likely resulted from Aβ accumulation and neuroinflammation. Stroke-operated mice retained cognition despite cortical thinning and hippocampal atrophy due to cerebrovascular adaptation, including increased CBF in the hippocampus and thalamus. Stroke did not worsen AD pathology, nor did AD exacerbate stroke outcomes. Sex differences were found: female APP/PS1 mice had more severe Aβ deposition, hyperactivity, lower body weight, and reduced CBF but less neuroinflammation, suggesting potential neuroprotection. These findings highlight white matter degeneration and Aβ pathology as key drivers of cognitive decline in AD, with stroke-related deficits mitigated by (cerebro)vascular adaptation. Sex-specific therapies are crucial for AD and stroke.

## 1. Introduction

Both ischemic stroke (IS) and Alzheimer’s disease (AD) cause an immense socioeconomic burden on global healthcare, with a lifetime risk of 50% for women and 30% for men of developing either condition [1]. Both diseases are leading causes of dementia [2]. Despite their partly different etiopathogeneses, AD and IS share several risk factors, such as age, sex, hypertension, and obesity [2,3]. Although the coexistence of AD and stroke and their potential additive impact on cognitive decline are acknowledged, a complete understanding of their pathophysiological overlap remains elusive [2,4].

Early-onset AD, caused by genetic mutations (APP, PSEN1, and PSEN2 genes), leads to increased amyloid-β (Aβ) production in brain areas critical for learning and memory [5]. However, late-onset AD is a multifactorial disease with various risk factors, including metabolic and vascular dysfunction, which affect patients years before clinical onset [6,7]. Autopsy studies have shown that 80% of all AD patients exhibit signs of small vessel disease (SVD), confirming cerebrovascular malfunction as an early key feature [8].

Vascular risk factors in AD are associated with a higher cerebral Aβ burden [9], increased chronic inflammation [5], and cerebral hypoperfusion [10], which synergistically promote neurodegeneration, leading to cognitive decline [11]. Importantly, epidemiological studies indicate a bidirectional relationship between AD and IS [12,13]. IS doubles the risk of AD [14] and vice versa [15]. In individuals with AD, Aβ can accumulate in the walls of cerebral blood vessels, termed cerebral amyloid angiopathy (CAA), causing blood–brain barrier (BBB) disruption and cerebral hypoperfusion [16]. Similarly, IS shares the same metabolic and vascular risk factors as AD, including SVD, and further contributes to decreased cerebral blood flow (CBF) [17] and increased neuroinflammation, both of which further aggravate vascular dysfunction and Aβ deposition [18]. Therefore, as AD and IS share multiple pathological mechanisms, with each condition exacerbating the other, their co-occurrence creates a self-perpetuating cycle of neurodegeneration.

Studies show that sex affects the prevalence, symptom progression, and severity of AD and IS differently. Men are more susceptible to IS at a younger age, whereas postmenopausal women are at higher risk and tend to suffer more severe ischemic damage later in life [19]. Premenopausal women have a low incidence of (cerebro)vascular diseases due to the cardiovascular protective impact of estrogen [20]. Notably, these sex-based differences extend to the co-occurrence of AD and IS, with women having a greater likelihood of cognitive impairment post-stroke [17]. Regarding AD, almost two-thirds of patients are women, not only because of their longer life expectancy but also because of other distinct biological (e.g., genetic predisposition) and psychological (e.g., depression, loneliness) risk factors that promote the progression of AD in women [21]. Notably, women with AD display poorer cognitive performance than men despite having similar levels of brain atrophy [22]. Yet, a significant knowledge gap exists in understanding how sex-specific disparities contribute to the divergent impacts of IS on cognitive impairment in AD, particularly in the context of their combined occurrence.

Thus, our study aims to address the critical research question: How does ischemic stroke affect male and female AD patients differently in terms of cognitive impairment? Therefore, we seek to elucidate the extent to which sex influences cognitive outcomes following a transient middle cerebral artery occlusion (tMCAo) in a male and female APP_swe_/PS1_dE9_ (APP/PS1) transgenic AD mouse model. We used magnetic resonance imaging (MRI) to assess temporal changes in CBF and gray matter (GM) and white matter (WM) integrity at 0.5, 4, and 8 months post-stroke, as well as cognition using the Morris water maze at 8 months post-stroke. Additionally, other in vivo physiological parameters were studied alongside post-mortem quantification of myelin density, neuroinflammation, and Aβ accumulation. Understanding the influence of sex on the manifestation, progression, and outcome of AD and stroke can guide the development of targeted, personalized treatments to improve outcomes for both men and women suffering from AD and IS.

## 2. Materials and Method

### 2.1. Animals

This study was conducted and reported according to the ARRIVE guidelines [23]. We performed a double-blinded study using 3-month-old male and female APP_swe_/PS1_dE9_ (APP/PS1) and C57BL/6JOlaHsd (WT) littermates. The APP/PS1 founder mice originated from the Johns Hopkins University, Baltimore, MD, USA (D. Borchelt and J. Jankowsky). All animals used in this study were bred at the Central Animal Facility, Radboud University Medical Center, Nijmegen, The Netherlands. This line was originally maintained on a hybrid background by backcrossing to C3HeJ×C57BL/6J F1 mice. For the present study, the breeder mice were backcrossed to C57BL/6J for fifteen generations. The sample size was estimated through a power analysis conducted prior to the start of the experiment (Appendix A). Mice were housed in digital ventilated cages (DVCs; Tecniplast SPA, Buguggiate (VA), Italy) containing corn-based bedding material (Bio Services, Uden, The Netherlands), wood wool nesting material (Bio Services, Uden, The Netherlands), and a mouse igloo (Plexx, Elst, The Netherlands) with access to conventional pellet food and tap water ad libitum. Prior to stroke induction, animals were group-housed by sex with a maximum of six animals per cage. Post-stroke, the mice were housed individually to avoid any potential fighting and to promote optimal wound healing. Temperature (21 °C ± 1 °C) and humidity (50–60%) in the animal facility were maintained at a constant level. An artificial 12 h light–dark cycle (lights on at 7 a.m.) was provided, and all behavioral and neuroimaging experiments were performed during daylight hours (7 a.m. to 6 p.m.) at the Preclinical Imaging Center (Radboud university medical center, Nijmegen, The Netherlands).

### 2.2. Study Design

This study focused on the longitudinal impact of stroke and Alzheimer’s disease on cognitive impairment, with a particular emphasis on sex differences. To this end, we included both 3-month-old male and female WT and APP/PS1 mice in this study. Initially, we measured baseline physiological parameters, including body weight and systolic blood pressure. At 3.5 months of age, the animals underwent either a right transient middle cerebral artery occlusion (tMCAO) or a sham surgery, resulting in eight experimental groups: (1) male WT sham, (2) male WT stroke, (3) male APP/PS1 sham, (4) male APP/PS1 stroke, (5) female WT sham, (6) female WT stroke, (7) female APP/PS1 sham, and (8) female APP/PS1 stroke.

Post-stroke, we monitored body weight and systolic blood pressure monthly. Neuroimaging was performed at 0.5, 4, and 8 months post-stroke. In DVCs, the walking patterns of each mouse were measured individually 24/7 for eight months. At 12 months of age, 8 months after stroke induction, we performed a Morris water maze test to analyze spatial learning and memory. After the last neuroimaging session, all mice were sacrificed, and the brains were collected for post-mortem analysis (immunohistochemical stainings, polarized light imaging, and biochemistry) (Figure 1). For analysis, only animals that completed the entire experiment were included. The initial and final group sizes, as well as the mortality rate, are specified in Appendix A.

### 2.3. Transient Middle Cerebral Artery Occlusion

At 4 months of age, animals were subjected to a transient right middle cerebral artery occlusion (tMCAO) to model an ischemic stroke as previously described [24]. Animals were anesthetized with 1.5–2% isoflurane (Abbott Animal Health, Abbott Park, IL, USA) in a 2:1 mixture of air and oxygen. Furthermore, all animals received a preoperative subcutaneous Rimadyl injection (5 mg/kg) for pain relief and to prevent infection. We performed a midline neck incision and inserted a 7–0 coated monofilament (190–200 μm, coating length 2–3 mm, 70SPRePK5, Doccol Corp., Sharon, MA, USA) into the right common carotid artery, pushing it upward along the right internal carotid artery, where it occluded the right middle cerebral artery (MCA) proximally in the Circle of Willis. Reperfusion of the MCA territory was achieved by withdrawing the filament after 30 min. Control mice underwent a similar procedure but with immediate reperfusion. Body temperature was maintained at 37 °C ± 0.5 °C using a heating pad. Cerebral blood flow of the MCA region was monitored with a Laser Doppler Flow probe (moorVMS-LDF2, Moor Instruments, Axminster, Devon, UK), which was fixed to the skull. A reduction in cerebral blood flow (>80%) confirmed successful stroke induction.

### 2.4. Systolic Blood Pressure: Tail-Cuff Plethysmography

Systolic blood pressure (SBP) was measured using a warmed tail-cuff plethysmography device (IITC Life Science Instruments, Woodland Hills, CA, USA). Baseline SBP was assessed prior to stroke induction, with monthly follow-up measurements taken after surgery. To avoid stress, the animals were habituated to the setup at 3.5 months of age. The mice were placed in appropriately sized, preheated Plexiglas restrainers equipped with tail cuffs and pulse sensors. The restrainers were then placed in a 38 °C warming chamber to maintain body temperature. SBP was measured over ten consecutive measurements using the software BPMonWin 1.35 (IITC Life Science Instruments, Woodland Hills, CA, USA). The average SBP (millimeters of mercury) of all measurements was calculated, excluding the first three measurements (habituation), as well as invalid measurements, such as those affected by motion artifacts.

### 2.5. Morris Water Maze

#### 2.5.1. Acquisition and Probe Trial

Spatial learning and memory were evaluated using the Morris water maze (MWM) test, conducted 8 months post-surgery at 12 months of age. The test was carried out in a 108 cm diameter circular pool filled with water made opaque with milk powder at 21 °C. A platform, 8 cm in diameter, was placed 1 cm below the surface in the northwest quadrant and surrounded by four visual cues for orientation. During the 4-day acquisition phase, mice were trained to find the hidden platform with four trials each day, separated by a maximum interval of one hour. We chose a longer intertrial interval than the standard to prevent stress-induced epileptic seizures in APP/PS1 mice and to allow stroke-operated mice with motor deficits more recovery time. Mice were released from different cardinal points (south, north, west, east) for each trial. If they failed to locate the platform within 120 s, they were guided to the platform and remained there for 30 s. A trial was considered successful if a mouse independently found the platform in less than 120 s. On the final day, a probe trial assessed spatial memory retention by removing the platform and allowing free swimming for 120 s. All trials were videotaped, and metrics such as escape latency, swim distance, swim velocity, and average distance to the platform were analyzed using EthoVision XT16 (Noldus, Wageningen, The Netherlands). To measure spatial memory, swim velocity, swim distance, average distance to the former platform, time spent in the former platform quadrant, and frequency of crossing the former platform location were assessed.

#### 2.5.2. Search Strategy Analysis

During acquisition and probe trials, search strategies were analyzed using Pathfinder (Jason Snyder Lab, Vancouver, Canada), as previously reported [25,26]. In total, eight distinct swimming strategies were identified: direct path, directed search, focal search, indirect search, chaining, scanning, random search, and thigmotaxis [25]. As previously described, mice progress through distinct search strategies reflecting their spatial learning. Initially, they exhibit thigmotaxis, swimming along the pool wall due to stress or unfamiliarity. As animals start exploring, they switch to random search, moving aimlessly in zig-zag or circular patterns. Afterward, they start scanning their environment, seeking landmarks without a clear goal. As learning progresses, animals adopt chaining, swimming in a circular pattern at a fixed distance from the wall, a non-spatial strategy based on external cues. Transitioning to indirect search, they move toward the platform with deviations, followed by focal search, where animals spend most of their time around the platform. Finally, they reach directed search, swimming with minimal detours toward the platform, and ultimately, the direct path, where they directly navigate toward the platform [25,26]. These strategies were defined based on specific spatial parameters, which were tailored to fit the experimental setup, including the platform position and diameter, maze diameter, maze center, angular corridor width, chaining annulus width, and thigmotaxis zone size [25].

A ‘direct path’ strategy is characterized by a swimming mouse taking almost the perfect route to the platform, with minimal deviation from a straight path. A ‘directed search’ strategy involves minor deviations from the direct route, while a ‘focal search’ refers to a spatially restricted search (in the center portion of the pool). In contrast, an ‘indirect search’ indicates a spatially targeted search that contains a major directional error. ‘Chaining’ describes a spatially non-specific strategy where mice search at a fixed distance from the pool wall. ‘Scanning’ refers to a random strategy that avoids the wall, whereas ‘random search’ does not include a spatial search pattern. ‘Thigmotaxis’ involves swimming close to, and often along, the pool wall.

These swimming strategies were classified into hippocampus-dependent (spatial) strategies, including direct path, directed search, focal search, and indirect search, and non-spatial strategies, including chaining, scanning, random search, and thigmotaxis. We analyzed the group differences in the percentage of hippocampus-independent (non-spatial) search strategies during the acquisition phase, but no significant differences were found (Appendix A).

#### 2.5.3. Cognitive Score

Cognitive performance was evaluated using a scoring system in which search strategies that are hippocampus-dependent receive higher scores: thigmotaxis = 0; random search = 1; scanning = 2; chaining = 3; indirect search = 4; focal search = 4; directed search = 5; direct path = 6 [25]. The average cognitive score was calculated for each mouse per day and normalized to six, the highest possible score.

### 2.6. Automated Locomotion and Trajectory Analysis

#### 2.6.1. Centroids

In this study, we utilized digital ventilated cages (DVCs) (Tecniplast S.p.A., Buguggiate (VA), Italy), which are equipped with sensing boards to continuously monitor the activity and location (centroids) of individually housed mice [27]. All system-specific details have been described previously [4]. These boards detect changes in electrical capacitance every 250 ms, reflecting the movement of animals within the cage due to the dielectric properties of matter affecting the proximity sensors. This allows for the recording of mice movements 24/7, capturing the position as x, y coordinates estimated from the weighted average position between active electrodes. For our analysis, we focused on weekend data to minimize interference from weekly activities (e.g., experiments and cage cleaning) and averaged the values for both day and night periods.

#### 2.6.2. Distance

Distance was calculated using the first and second derivatives of the centroid coordinates with respect to time, respectively. The distance was computed using Traja, as described elsewhere [28].

### 2.7. MRI Protocol

All animals were scanned at 4.5, 8, and 12 months of age, specifically at 0.5, 4, and 8 months after surgery, using an 11.7 T BioSpec Avance III small animal MR system (Bruker BioSpin, Ettlingen, Germany). This scanner is equipped for rodent imaging and includes an actively shielded gradient set with a strength of 600 mT/m operated using Paravision 6.0.1 software. For signal transmission and reception, a circular polarized volume resonator and an actively decoupled mouse brain quadrature surface coil were used, respectively (Bruker BioSpin, Ettlingen, Germany). During the MRI scans, the anesthetized mice were fixed in a stereotactic holder to prevent movement. Anesthesia was induced with 3.5% isoflurane in an oxygen–air mixture (1:2 ratio). The isoflurane level was reduced to approximately 1.5% for maintenance, based on the mice’s respiratory rate (approximately 100 bpm). Respiration was monitored with a pneumatic cushion respiratory monitoring system (Small Animal Instruments Inc., Stony Brook, NY, USA). In addition, a rectal probe monitored body temperature, which was maintained at physiological levels (approximately 37 °C) by a heated airflow. Following standard adjustments, an anatomical reference scan of the mouse brain was performed by acquiring gradient images in three orthogonal directions (axial, sagittal, and coronal). Imaging sequence details are provided in Appendix A.

#### 2.7.1. Hippocampal Volume and Cortical Thickness

Hippocampal volume and cortical thickness were analyzed using ImageJ (version 1.51, National Institutes of Health, United States) based on coronal T2-weighted images. For cortical thickness measurements, three cortical areas, the somatosensory cortex (bregma −0.94), auditory cortex (bregma −2.46), and visual cortex (bregma −2.46), were manually selected in both hemispheres using the mouse brain atlas by Franklin and Paxinos [29]. The average cortical thickness across all measurements was calculated.

Hippocampal volume was manually segmented from five consecutive coronal T2-weighted image slices in both the left and right hemispheres, delineating boundaries between bregma levels −0.94 and −3.40, as described in the mouse brain atlas by Franklin and Paxinos [29]. In cases where more than five slices were available, we prioritized the most frontal slices. For the right, stroke-affected hippocampus, only slices corresponding to those selected for the left hemisphere were included in the selection. Total hippocampal volume was calculated by summing the segmented areas across all selected slices and multiplying by the slice thickness.

Initially, one blinded researcher manually selected cortical thickness and hippocampal volumes for the entire dataset. To validate the measurements, a second researcher independently reviewed a random 10% subset of the data for accuracy.

#### 2.7.2. Cerebral Blood Flow

MR perfusion data were obtained using FAIR MRI techniques, derived from a series of echo planar images (EPIs) in three distinct regions of interest (ROIs) located at bregma −1.94, including the cortex, hippocampus, and thalamus, as per the Paxinos and Franklin mouse brain atlas [29]. Regional CBF was then computed as previously described [30]. In a subset of our MRI scans, we encountered technical issues due to damage to the MRI receiver, resulting in low cerebral blood flow (CBF) data. To objectively identify the affected scans, we computed the signal-to-noise ratio (SNR) for each scan, serving as an indicator of potential data inaccuracies. Once the scans with suboptimal SNR were identified, we conducted a correction procedure by first determining the average CBF for all ROIs across both valid (unaffected) and invalid (affected) scans. Subsequently, we calculated a correction factor, representing the ratio necessary to adjust the CBF values of the invalid scans to match the average CBF of the ROIs in the valid scans. Finally, this correction factor was applied to the CBF values of the invalid scans, thereby normalizing their data to match those of the valid scans.

#### 2.7.3. Diffusion Tensor Imaging

Gray matter (GM) and white matter (WM) integrity were assessed using diffusion tensor imaging, following established methodologies as described in previous studies [31,32]. In this study, we utilized fractional anisotropy (FA) and mean water diffusivity (MD) to quantify myelination and fiber density in WM and to evaluate membrane density in GM by MD [33,34,35,36]. FA and MD were not only measured across general GM and WM areas but also within manually identified ROIs, including the cortex, corpus callosum, fornix, and hippocampus, according to the mouse brain atlas of Paxinos and Franklin [29].

### 2.8. Tissue Preparation

At 12 months of age, mice were sacrificed by transcardial perfusion using 0.1 M phosphate-buffered saline (PBS, at room temperature 20–25 °C), and their brains were carefully extracted. The frontal parts of both brain hemispheres (bregma: 5.35 to 1.65) were snap-frozen, while the remaining brain tissue (bregma: 1.65 to −4) was postfixed in 4% paraformaldehyde for 24 h. Subsequently, the brain tissue was preserved in a 0.1 M PBS solution containing 0.01% sodium azide and stored at 4 °C until sectioning. Approximately 24 h before sectioning, using a microtome (Microm HC 440, Walldorf, Germany), the brains were cryoprotected by immersion in 30% sucrose dissolved in 0.1 M phosphate buffer. According to the atlas of Franklin and Paxinos [29], we cut eight series of 30 μm frontal sections, ranging from bregma 1 to −4. These sections were stored in a solution of 0.1 M PBS with 0.01% sodium azide in preparation for further analysis by immunohistochemical stainings and polarized light imaging (PLI).

### 2.9. Immunohistochemistry

A standardized protocol for 3,3′-diaminobenzidine-nickel (DAB-Ni) immunohistochemical staining was carried out as previously described [37]. Before incubation with primary antibodies, sections underwent pre-incubation for 30 min in a blocking solution comprising 3% BSA and 0.5% Triton X-100 in PBS. Two immunohistochemical stainings were conducted.

To assess the severity of neuroinflammation, activated microglia and macrophages were visualized using a staining against ionized calcium-binding adapter molecule 1 (IBA-1). The deposition of Aβ was visualized using the antibody WO-2 [38]. One series of free-floating sections was incubated with primary antibodies: (1) goat anti-IBA-1 (1:4000; Abcam, Cambridge, UK) and (2) WO-2: mouse anti-human Aβ4-10 (1:10.000; Centre for Molecular Biology, University of Heidelberg, Germany). Subsequently, secondary antibodies were added: (1) IBA-1 donkey anti-goat (1:1500; Jackson ImmunoResearch, West Grove, PA, USA), and (2) WO-2: donkey anti-mouse biotin (1:1500; Jackson ImmunoResearch, West Grove, PA, USA). Brain sections were mounted on gelatin-coated slides and dried at 37 °C overnight.

### 2.10. Quantification

Representative images were taken with the Axio Imager A2 (Zeiss, Germany) at 5× magnification. Depending on the availability of brain tissue, 1–2 brain sections proximal to bregma −1.94 were chosen, according to the Franklin and Paxinos brain atlas [29]. ROIs (bregma −1.94: cortex, hippocampus, and thalamus) were selected in ImageJ using the freehand tool in a double-blinded manner by two researchers. ROI definitions were finalized for further analysis upon agreement between both researchers. An intensity-based threshold was set in ImageJ to distinguish the target staining from non-specific background staining. IBA-1 density measurements were quantified as the relative IBA-1+ area per ROI.

### 2.11. Polarized Light Imaging

Polarized light imaging (PLI) was performed to analyze the fiber density and orientation in brain tissues, utilizing the birefringent properties of myelin as described in prior studies [39,40].

One series of 30 µm thick coronal brain sections was mounted on uncoated glass slides and coverslipped with polyvinylpyrrolidone mounting medium. Raw PLI images were taken with an Axio HV microscope (Zeiss, Germany) equipped with an RGB camera (AxioCam ERc 5s, Zeiss, Germany), a rotating polarizer, a quarter-wave plate, a stationary polarizer, and a white LED light source, as previously described [41,42].

We captured a dataset of raw PLI images comprising nine consecutive shots at rotation angles ranging from 0° to 160°. Background correction images were similarly taken in each imaging session. Subsequently, post-processing of the raw images was performed using Matlab R2018b (MATLAB R2018b; MathWorks Inc., Natick, MA, USA), where the images were fitted to the Jones formula to derive PLI maps. These maps included (1) transmittance, (2) retardance, (3) in-plane, (4) inclination, (5) dispersion, and (6) FOM-HSV values [39,43].

Myelin density (retardance) and orientation (dispersion) were quantified with ImageJ. Proximal to bregma −1.94, one slice was selected based on the mouse brain atlas of Franklin and Paxinos [29]. Two double-blinded researchers manually selected the cortex, corpus callosum, hippocampus, and thalamus using the freehand tool in ImageJ. The different ROIs were defined for further analysis when both researchers reached an agreement. The mean gray values of the retardance and dispersion maps are represented as percentages.

### 2.12. Aβ ELISA

Total protein was extracted from the snap-frozen left and right frontal cortex (bregma: 5.35 to 1.65) of APP/PS1 mice to determine Aβ_1–42_ to Aβ_1–40_ ratios, as described previously [40]. Enzyme-linked immunosorbent assay (ELISA) kits specifically for human Aβ_1–42_ (Invitrogen, Frederick, MD, USA; catalog KHB3441) and Aβ_1–40_ (Invitrogen, Frederick, MD, USA; catalog KHB3481) were used to quantify soluble and insoluble Aβ species according to the manufacturer’s instructions. Briefly, 25 ng of total protein extracts were adjusted to a final volume of 50 μL with the reagent standard and sample diluent supplied by the kit. Results are expressed as picograms (pg) per milligram (mg) of tissue. The mean of duplicate measurements is presented. No surgery- or sex-related differences were found (Appendix A).

### 2.13. Statistics

Mixed-design analysis of variance (ANOVA), a common statistical approach for analyzing between-subject differences in repeated measures datasets, does not accommodate missing values. In our longitudinal study design, subject attrition over time results in significant information loss. To address this issue, we employed linear mixed models (LMMs), which circumvent the problem of listwise deletion in the presence of missing data. The statistical analysis was performed using IBM SPSS Statistics 29 (IBM Corporation, Armonk, NY, USA), and results are expressed as means ± standard error of the mean (SEM).

More specifically, we performed an LMM analysis, first using a full model that included all fixed factors of interest and their interactions, including sex, genotype, surgery, time, and all possible two-, three-, and four-way interactions among these variables. Non-significant factors and interactions were subsequently excluded from the refined model. Finally, we conducted pairwise comparisons for each significant fixed factor or interaction identified. For longitudinal data, we applied Bonferroni multiple comparison correction.

Furthermore, for experiments conducted at single time points, such as the Morris water maze and post-mortem analysis, we also applied an LMM. The model included the fixed factors of sex, genotype, and surgery, as well as interhemispheric differences, particularly between the right (stroke-affected) and the left (stroke-unaffected) hemispheres, along with all possible two-, three-, and four-way interactions among these factors. Here, the Benjamini–Hochberg procedure was used to correct for multiple comparisons in datasets collected at a single time point.

In addition, a chi-square test was used for statistical analysis of categorical data (e.g., cognitive score).

A detailed overview of statistical analysis of all parameters is provided in the Appendix A. Graphs in the main manuscript illustrate the statistical effects of fixed factors (sex, genotype, surgery, time, left vs. right hemisphere) and their respective interactions, while raw data visualizations for each experimental group are included in the Appendix A for transparency.

## 3. Results

### 3.1. Body Weight

In this study, we tracked the body weight changes of mice across eight experimental groups, including both male and female WT and APP/PS1 mice after ischemic stroke induction. Body weight was measured pre-stroke (baseline) and subsequently on a monthly basis over an 8-month period following stroke surgery. Before stroke surgery (baseline), at 3.5 months of age, a clear sex effect was observed. Female mice weighed less than male mice, while no other differences were detected at baseline (Figure 2A, indicated with a).

Linear mixed model analysis revealed significant interactions between genotype, sex, and time. Male WT (between months: 1→2, 2→3, 3→4, 5→6, 1→8) and male APP/PS1 mice (between months: 1→2, 2→3, 3→4, 5→6, 1→8) significantly gained weight during 8 months after surgery (Figure 2A). Female WT and female APP/PS1 mice, however, displayed less pronounced weight gain (female WT between months: 1→2, 5→6, 1→8, female APP/PS1 between months: 1→2, 5→6, 1→8) over time and maintained lower body weight than their male counterparts throughout the 8 months post-surgery (months 1–8) (Figure 2A, indicated with b, c). From 4 months post-surgery onward, male APP/PS1 mice were heavier than male WT mice (months 4–8) (Figure 2A, indicated with d), while among the females, APP/PS1 mice had a lower body weight than WT mice, reaching significance in the final two months of the study (months 7–8) (Figure 2A, indicated with e).

Additionally, in the follow-up months, significant interactions involving genotype, surgery, and time in relation to body weight changes were observed. Both WT and APP/PS1 mice, irrespective of their sex, exhibited differences in body weight, albeit with distinct patterns (Figure 2B). A body weight gain was noted in WT sham (between months: 1→2, 2→3, 3→4, 5→6, 1→8), WT stroke (between months: 1→2, 2→3, 1→8), APP/PS1 sham (between months: 1→2, 1→8), and APP/PS1 stroke mice (between months: 1→2, 2→3, 5→6, 1→8) during the post-surgery period (Figure 2B). Notably, from 6 months post-surgery onward, WT mice subjected to stroke consistently displayed a reduction in body weight compared to the control group that underwent sham surgery (months 6–8) (Figure 2B, indicated with f). In contrast, APP/PS1 mice did not exhibit stroke-related weight differences with age (Figure 2B).

### 3.2. Systolic Blood Pressure

We measured systolic blood pressure (SBP) at baseline (at 3.5 months of age) and monthly after stroke induction. At baseline, APP/PS1 mice displayed higher SBP than their WT littermates (Figure 2C, indicated with g). Post-surgery, APP/PS1 mice maintained higher SBP compared to WT animals. The consistently higher SBP in APP/PS1 mice, both at a young age and longitudinally post-surgery, highlights the potential impact of Alzheimer’s disease pathology on the cardiovascular system.

### 3.3. Digital Ventilated Cages—Distance

All mice were single-housed for 8 months post-surgery in digital ventilated cages that meticulously monitored activity and walking patterns. As nocturnal animals, all mice walked longer distances at night.

In the light phase, overall sex, genotype, and surgery effects were detected: female mice walked more than males (Figure 2D), stroke-operated mice walked more than sham-operated mice (Figure 2E), and APP/PS1 mice walked more than WT mice (Figure 2F).

Similarly, during the dark phase, stroke-operated mice walked consistently larger distances than sham-operated mice (Figure 2H). A significant sex and genotype interaction was observed over time. Both, among WT (at months 2, 3, 4, 6, and 8 post-surgery) (Figure 2G, indicated with b) and APP/PS1 animals (all 8 months post-surgery) (Figure 2G, indicated with c), female mice walked more than male mice. Additionally, APP/PS1 mice walked longer distances than WT mice among males (1, 2, 3, 7, and 8 months post-surgery) (Figure 2G, indicated with d) and females (all 8 months post-surgery) (Figure 2G, indicated with e).

### 3.4. Morris Water Maze

#### 3.4.1. Acquisition

In the Morris water maze, performed 8 months post-surgery at 12 months of age, we tested spatial learning and hippocampal-dependent memory in both male and female, sham- or stroke-induced APP/PS1 and WT mice. Over the course of 4 acquisition days, we observed a consistent decrease in the time required to find the hidden platform, indicating the ability of effective spatial learning across all groups (acquisition days: 1→2, 2→3, 3→4, 1→4) (Appendix A). In line with this, the swimming distance (acquisition days: 1→2, 2→3, 1→4) (Appendix A) and the average distance to the hidden platform (acquisition days: 1→2, 2→3, 1→4) (Appendix A) also declined, further supporting a learning curve in spatial memory. When comparing the APP/PS1 mice to the WT group, we found several notable differences. APP/PS1 mice exhibited a higher latency to find the platform, which is in accordance with the expected AD-related decline in spatial learning (Figure 3A). This was further supported by the observation that APP/PS1 mice also had longer swimming distances compared to the WT mice (Figure 3C). Notably, female mice demonstrated longer swimming distances than male mice regardless of their genotype and type of surgery (Figure 3D). An interaction of sex, genotype, and time was present when analyzing swimming velocity. Specifically, male WT mice showed a decrease in swimming velocity over time (between acquisition days: 2→3, 1→4) (Figure 3E). On days 3 and 4, female WT mice outperformed their male counterparts in swimming velocity (Figure 3E, indicated with b). Furthermore, on day 4, male APP/PS1 mice swam faster than their WT counterparts, indicating a distinct influence of genotype on swimming behavior (Figure 3E, indicated with d).

#### 3.4.2. Probe

Throughout the probe trial (2 min), female mice swam at a lower velocity than male mice (Appendix A). During the first 30 s of the trial, APP/PS1 mice, on average, maintained a greater distance from the former platform location compared to WT mice, indicative of a diminished spatial memory associated with AD (Figure 3H). Additionally, we observed an interaction between sex and surgery type regarding swim velocity and swim distance. Specifically, female mice that underwent stroke surgery swam shorter distances (Figure 3F) and at a slower pace (Figure 3G) than their male counterparts. In contrast, male mice subjected to stroke surgery exhibited increased swim distances (Figure 3F) and velocity (Figure 3G) compared to males who underwent sham surgery.

#### 3.4.3. Cognitive Score

The cognitive level of a search strategy can be measured with a cognitive score that evaluates swim strategies based on their relevance to spatial learning—a navigation strategy dependent on hippocampal function [25]. Higher scores indicate strategies focused on spatial learning, while lower scores are associated with non-spatial strategies like ‘random search’, ‘scanning’, and ‘chaining’. The cognitive scores of both WT and APP/PS1 mice increased during the acquisition training; however, APP/PS1 animals displayed overall lower cognitive scores than WT mice (Figure 3B).

### 3.5. Cortical Thickness

In the present study, we assessed the cortical thickness using MRI scans in all experimental groups at 0.5, 4, and 8 months after surgery. At all timepoints post-surgery, cortical thickness was lower in the stroke-affected (ipsilateral) hemisphere compared to the control (contralateral) hemisphere (Figure 4A). The ipsilateral cortex was, in fact, overall thinner in stroke-operated compared to sham-operated animals (Figure 4B). Additionally, we found consistent sex and genotype effects on cortical thickness in both hemispheres throughout the study. In females, the contralateral (Appendix A) and ipsilateral (Figure 4C) cortex was overall thicker than in male mice. Particularly, at 0.5 and 4 months post-stroke, this sex effect on cortical thickness reached significance (Figure 4D). Regarding genotype effects, APP/PS1 mice demonstrated a thicker cortex compared to their WT counterparts in both the contralateral (Appendix A) and ipsilateral hemisphere (Figure 4E) over time, with significant differences emerging at later timepoints, particularly at 4 and 8 months post-surgery (Figure 4F).

### 3.6. Hippocampal Volume

In the present study, we assessed hippocampal volume using neuroimaging at 0.5, 4, and 8 months after surgery across all experimental groups. Major effects of the stroke surgery were observed. The ipsilateral hippocampal volume was consistently lower in stroke-operated mice compared to sham-operated mice across all time points (Figure 4G,H). Furthermore, at all timepoints, stroke-operated mice exhibited lower hippocampal volume in the ipsilateral hemisphere compared to the contralateral hemisphere (Figure 4H). At 4 months after surgery, sham mice also demonstrated similar interhemispheric differences (Figure 4H). In addition to the stroke effect, we also found that in females, the APP/PS1 group had a larger contralateral hippocampal volume than WT mice (Appendix A).

### 3.7. Cerebral Blood Flow

Cerebral blood flow (CBF) was assessed through neuroimaging at 0.5, 4, and 8 months following stroke induction. We analyzed regions of interests (ROIs) within the brain, including the cortex, hippocampus, and thalamus, in both the ipsilateral (stroke-affected) and contralateral (control) hemispheres. To assess CBF differences between the experimental groups, we used a linear mixed model for a detailed analysis. This model was used to separately assess how sex, surgery type, genotype, and time, as well as their interactions, influenced CBF in each hemisphere longitudinally. In addition, we performed separate analyses for each MRI time point to investigate the effect and interactions of interhemispheric differences, sex, surgery type, and genotype on CBF.

#### 3.7.1. Cortex

At 0.5, 4, and 8 months post-surgery, all experimental groups, regardless of undergoing sham or stroke surgery, had a lower CBF in the ipsilateral cortex in comparison to the corresponding contralateral cortical region (Figure 5A). Additionally, we found a sex difference in cortical CBF over time in both hemispheres, with female mice showing a lower CBF than male mice (Figure 5B and Appendix A). This sex difference became significant, particularly at 0.5 and 8 months after surgery (Figure 5C). Moreover, at 4 months after surgery, we detected a genotype effect, with APP/PS1 mice exhibiting lower CBF compared to WT mice (Figure 5D).

#### 3.7.2. Hippocampus

Similar to the cortex, in the hippocampus, we found an overall lower CBF in the ipsilateral hemisphere compared to the contralateral hemisphere across all experimental groups, regardless of the surgery type, at 0.5, 4, and 8 months post-stroke (Figure 5E). Notably, when the hippocampal CBF of the contralateral and ipsilateral hemispheres were examined together at 4 months post-stroke, the stroke groups had higher CBF than their sham-operated littermates (Figure 5F). However, only the contralateral hippocampus demonstrated a reduction in CBF changes over time, with decreasing CBF from 0.5 to 8 months post-stroke, while no CBF change was observed in the ipsilateral, stroke-affected hippocampus (Appendix A). A sex effect in the hippocampus was present shortly after surgery (0.5 months), with female mice exhibiting lower CBF than male mice (Figure 5G). At 8 months post-surgery, a sex genotype interaction was detected. Female APP/PS1 mice had lower hippocampal CBF compared to male APP/PS1 mice, and male APP/PS1 mice exhibited higher CBF than male WT mice (Figure 5H).

#### 3.7.3. Thalamus

In the ipsilateral thalamus, CBF was higher in stroke-operated animals compared to their sham-operated littermates throughout the post-surgery period (Figure 5I). In comparison, in the contralateral thalamus, surgery effects were more nuanced, revealing an interaction between surgery and genotype (Appendix A). Specifically, in the contralateral thalamus, only APP/PS1 stroke mice exhibited higher CBF than APP/PS1 sham mice, and APP/PS1 sham mice had overall lower CBF in comparison to WT sham mice (Appendix A). Regarding interhemispheric differences, we found that the ipsilateral thalamus showed lower CBF than the contralateral at 0.5 months after surgery, regardless of the surgery type (Figure 5K). However, in particular, at 4 and 8 months post-surgery, stroke mice exhibited higher thalamic CBF than their sham-operated counterparts (Figure 5J). Similar to the cortex, an overall sex difference in thalamic CBF over time was observed in the contralateral hemisphere, with female mice exhibiting lower CBF than male mice (Appendix A). In the combined thalamus, this sex effect was observed 8 months after surgery (Figure 5L). Additionally, in the ipsilateral hemisphere, this sex effect became specifically evident within the genotypes (Figure 5M). At 0.5 months post-stroke induction, female APP/PS1 mice showed lower CBF than male APP/PS1 mice (Figure 5M, indicated with c), and 4 months after surgery, female WT had lower CBF than male WT mice (Figure 5M, indicated with b). Moreover, at 4 months post-stroke, thalamic CBF was lower in male APP/PS1 mice compared to male WT mice (Figure 5M, indicated with d). Notably, an increase in CBF was observed in the ipsilateral thalamus between 0.5 and 4 months after surgery in male WT and female APP/PS1 mice (Figure 5M).

Figure 6 shows voxel-wise CBF images in male and female WT and APP/PS1 mice at 0.5, 4, and 8 months post-stroke or -sham surgery.

### 3.8. Diffusion Tensor Imaging

In this study, we longitudinally assessed gray matter (GM) and white matter (WM) integrity at 0.5, 4, and 8 months post-stroke using in vivo diffusion tensor imaging (DTI). We analyzed fractional anisotropy (FA) to evaluate myelination and fiber density, as well as mean diffusivity (MD) as an inverse measure of membrane density. WM degeneration after ischemic stroke is associated with reduced FA and increased MD. We analyzed the FA and MD of total WM, GM, corpus callosum, fornix, and the cortex and hippocampus in the ipsilateral (stroke-affected) and contralateral (stroke-unaffected, control) hemispheres, respectively.

### 3.9. Fractional Anisotropy

Early in the postoperative period, from 0.5 to 4 months post-stroke, we observed an increase in FA in the GM, corpus callosum, and fornix, indicating recovery of myelin and fiber density within 4 months post-stroke (Appendix A). However, with advancing age and progressing AD pathology, from 4 to 8 months post-surgery, FA decreased in the WM, GM, corpus callosum, and fornix across all experimental groups (Appendix A).

Focusing on the cortex, at 4 months post-surgery, the ipsilateral hemisphere showed higher cortical FA than the contralateral cortex, regardless of sex, surgery type, or genotype of the mice (Figure 7A). No interhemispheric differences were present at 8 months post-stroke as FA decreased in the ipsilateral cortex between 4 and 8 months after surgery (Appendix A). Additionally, we found a genotype–surgery interaction in the cortex (Figure 7B). Four months after surgery, APP/PS1 sham mice showed lower FA than WT sham animals, while stroke-operated APP/PS1 animals displayed higher FA than APP/PS1 sham mice (Figure 7B). This effect was driven by a drop in FA in the contralateral cortex of APP/PS1 sham mice between 0.5 and 4 months after surgery, resulting in lower FA in sham-operated APP/PS1 mice compared to their stroke-operated littermates, particularly in the contralateral cortex (Appendix A, indicated with h). Moreover, by 8 months after stroke induction, FA in the contralateral cortex of WT stroke mice was significantly higher than at 0.5 and 4 months after surgery (Appendix A). This increase in FA levels resulted in higher FA in the contralateral cortex of WT stroke mice in comparison to WT sham mice (Appendix A, indicated with f). Additionally, APP/PS1 stroke mice exhibited lower FA in the contralateral cortex compared to WT stroke mice (Appendix A, indicated with i). A genotype effect was also detected when analyzing the combined cortex, with APP/PS1 mice demonstrating lower FA than WT mice 8 months post-surgery (Figure 7C).

Regarding sex-specific differences in the cortex, at 8 months post-surgery, we found that female mice showed lower FA than male mice (Figure 7D), a difference that was also observed in the contralateral cortex (Appendix A, indicated with a). In male mice, particularly, we detected an increase in FA in the contralateral cortex from 4 to 8 months post-surgery (Appendix A).

Throughout the entire postoperative period, FA in the ipsilateral hippocampus was consistently lower than in the contralateral, regardless of sex, surgery type, or genotype (Figure 7E). Notably, there was a progressive increase in FA in the ipsilateral hippocampus from 0.5 to 4 months, as well as in the contralateral and ipsilateral hippocampus from 0.5 to 8 months post-surgery (Appendix A). At both 4 and 8 months after surgery, hippocampal FA was lower in APP/PS1 mice compared to WT mice (Figure 7F). This genotype effect was particularly evident in the contralateral hippocampus over time (Appendix A).

### 3.10. Mean Diffusivity

The analysis of MD, a marker for neurodegeneration, edema, and necrosis in GM after an ischemic stroke, revealed no significant changes in GM and WM MD levels (Appendix A). However, in the corpus callosum and fornix WM tracts, we found a decrease in MD between 0.5 and 8 months post-stroke across all groups (Appendix A).

Moreover, both the contralateral and ipsilateral cortex showed a decrease in MD between 0.5 and 4 months post-stroke (Appendix A). Across all measurement time points, the ipsilateral cortex exhibited higher MD values than the contralateral cortex (Figure 7G).

In the hippocampus, we did not observe any significant changes in MD over time, neither in the contralateral nor in the ipsilateral hemisphere. However, when analyzing each time point individually, several effects became evident. At 0.5 months after stroke, the ipsilateral hippocampus exhibited higher MD compared to the contralateral hippocampus (Figure 7H). At the same time point, stroke mice displayed lower MD than sham-operated mice (Figure 7I). Additionally, a genotype–sex interaction was identified in the hippocampus at 0.5 months post-surgery: female WT mice exhibited lower MD compared to male WT mice, while female APP/PS1 mice exhibited higher MD than female WT mice (Figure 7J).

### 3.11. Polarized Light Imaging

In addition to DTI, we performed polarized light imaging (PLI) to quantify myelin density and orientation in several regions of interest post-mortem, including the cortex, corpus callosum, hippocampus, thalamus, and fornix.

#### 3.11.1. Retardance

PLI retardance values represent the degree of myelination, with decreased retardance values being an indirect measure of myelin loss. In the cortex, male APP/PS1 mice exhibited higher myelin density than their WT counterparts, suggesting better myelination (Figure 8A). Interestingly, in female mice, the opposite genotype effect was observed; female APP/PS1 mice showed a reduction in myelin density compared to WT females (Figure 8A). Moreover, these female APP/PS1 mice had lower cortical myelin density than their male counterparts, highlighting a sex difference within the APP/PS1 genotype (Figure 8A).

A consistent genotype effect was observed in both the hippocampus (Figure 8B) and thalamus (Figure 8C): APP/PS1 mice, irrespective of sex and surgery type, showed reduced myelin density in these ROIs compared to WT mice. This genotype effect was particularly pronounced in the contralateral hemisphere of the hippocampus (Appendix A). Within the combined thalamus (Figure 8D), as well as in the contralateral and ipsilateral thalamus separately (Appendix A), female mice displayed higher myelin density than males, suggesting a potential protective effect against myelin loss in female brains.

#### 3.11.2. Dispersion

Dispersion levels are a quantitative estimate of fiber orientation. Lower dispersion values indicate better myelin quality [40]. In the cortex of APP/PS1 mice, females had lower dispersion than males, indicating better myelin quality (Figure 8F). Within female mice, APP/PS1 mice displayed lower dispersion, hence better myelin quality, compared to WT animals (Figure 8F).

A complex interaction was observed in the contralateral part of the corpus callosum (Appendix A). Here, female stroke-operated WT and female sham-operated APP/PS1 mice demonstrated lower dispersion levels than their male counterparts, implying better myelin quality among male animals (Appendix A). An overall sex difference was also present in the entire corpus callosum: female mice exhibited lower dispersion than males (Figure 8G).

Contrary to the cortex and corpus callosum findings, an overall higher dispersion was observed in the thalamus of female animals compared to males (Figure 8H), particularly in the contralateral thalamus (Appendix A). This suggests a difference in myelin quality between different brain regions, with females showing lower myelin quality in the thalamus. Additionally, in the thalamus, sham mice displayed lower dispersion than stroke-operated mice, suggesting a stroke-related reduction in myelin quality (Figure 8I). 

### 3.12. Neuroinflammation

Neuroinflammation was evaluated post-mortem via immunohistochemistry in male and female, sham- or stroke-operated WT and APP/PS1 mice, 8 months post-surgery at 12 months of age. Ionized calcium-binding adapter molecule 1 (IBA-1) is a marker specifically expressed in activated microglia and macrophages, providing insight into inflammatory responses in the brain. We assessed the number of IBA-1+ cells and the relative area of IBA-1+ staining in the brain.

In the cortex, female WT animals exhibited fewer activated microglia compared to male WT animals (Figure 9A). However, both male and female APP/PS1 mice demonstrated a higher count of activated microglia than their WT counterparts (Figure 9A). The same genotype effect was observed consistently across both hemispheres (Figure 9B). Additionally, APP/PS1 mice showed a larger area of IBA-1+ staining in the combined cortex and in both hemispheres individually, indicating more extensive neuroinflammation compared to WT mice (Appendix A).

The hippocampus displayed similar patterns, with female mice showing fewer activated microglia than males (Figure 9C), particularly in the contralateral hippocampus, where this sex effect was more pronounced in stroke-operated animals (Figure 9D). Furthermore, female mice also displayed a smaller IBA-1+ area than males in the ipsilateral hippocampus (Appendix A). Interestingly, this sex difference was particularly detectable in the stroke-operated mice: female stroke mice had a smaller IBA-1+ area than male stroke mice in the hippocampus (Appendix A). Furthermore, stroke-operated male mice showed a larger IBA-1+ area compared to sham-operated male animals, while such a stroke effect was not present in the hippocampus of female mice (Appendix A). APP/PS1 mice, regardless of sex and surgery type, had a higher amount of activated microglia in both the combined (Figure 9E) and contralateral hippocampus (Figure 9F), as well as a larger IBA-1+ area in the combined (Appendix A) and contralateral hippocampus (Appendix A) compared to WT mice.

The impact of stroke surgery was clearly visible in the thalamus. Among stroke mice, APP/PS1 animals had significantly less activated microglia (Figure 9G) and a smaller IBA-1+ area in the thalamus compared to WT mice (Appendix A). Notably, stroke-operated APP/PS1 mice exhibited a lower amount of microglia than their sham-operated counterparts (Figure 9G). Concerning the IBA-1+ area, stroke-operated WT mice showed an increased IBA-1+ area compared to sham-operated WT mice (Appendix A).

### 3.13. Amyloid Beta

Amyloid beta deposition is a key hallmark of AD. It was assessed in immunohistochemically stained samples using the WO-2 antibody in male and female, sham- or stroke-operated WT and APP/PS1 mice, 8 months post-surgery at 12 months of age. The relative area covered with Aβ deposition was assessed in the combined hippocampus. The analysis revealed a significant sex effect. Female APP/PS1 mice showed a larger relative Aβ+ area compared to the male APP/PS1 mice (Figure 10A).

## 4. Discussion

Epidemiological studies have shown a reciprocal relationship between ischemic stroke (IS) and Alzheimer’s disease (AD), as they often manifest together and affect each other. Both conditions can cause severe cognitive deficits [2]. Given the influence of sex hormones on cognitive decline and neurodegeneration, investigating sex disparities in IS and AD is critical, particularly as estrogen provides neuroprotection in women [20,44]. Our study aimed to investigate sex-specific differences after IS in a mouse model of AD using APP_swe_/PS1_dE9_ (APP/PS1) transgenic mice.

This study identified several important findings. APP/PS1 mice exhibited impairments in spatial learning and memory, hyperactive behavior, hypertension, white matter (WM) degeneration, increased cortical thickness, and chronic neuroinflammation.

Stroke reduced hippocampal volume and cortical thickness in the ipsilateral hemisphere, as well as temporary regional hyperperfusion. Both sham and stroke surgeries caused longitudinal hypoperfusion and diminished GM integrity. Stroke in particular caused cortical thinning, hippocampal atrophy, and increased mean. Post-mortem, stroke mice had reduced myelin fiber quality without evident myelin loss. WM initially recovered but degenerated in all groups long-term post-surgery. Despite marked structural deterioration, cognition remained intact in stroke mice.

Female mice exhibited lower CBF and higher Aβ burden, implicating greater susceptibility to Aβ-related vascular dysfunction. Female APP/PS1 mice showed increased cortical thickness and hippocampal volumes, possibly due to early Aβ deposition. They also showed more hyperactivity, reflecting severe motor symptoms in female AD patients, but less neuroinflammation, possibly due to the neuroprotective effects of estrogen.

### 4.1. Cognition and Locomotion

At 12 months of age, all mice showed spatial learning abilities in the Morris water maze (MWM), but learning abilities were impaired in APP/PS1 mice, which took longer to find the hidden platform and swam larger distances than WT mice. In accordance with the literature, spatial learning deficits in the MWM are reported in APP/PS1 as early as 6 months of age and become more pronounced by 12 months of age [45,46]. Cognitive scores, based on the search strategies used, indicate that APP/PS1 mice relied more on non-spatial, hippocampus-independent strategies rather than spatial, hippocampus-dependent strategies. Additionally, APP/PS1 mice showed impaired spatial memory, as indicated by a higher average distance swum to the platform during the probe trial. These results align well with the known cognitive deficits associated with AD [2]. Contrary to the literature, no stroke-induced cognitive decline was found [47], supposedly because the MWM test was performed 8 months after stroke induction. The cortex and hippocampus have the capacity for structural and functional recovery of brain networks after an ischemic event, especially in young and middle-aged mice [47,48,49].

Note that during the probe trial, stroke-operated female mice displayed lower swim distance and velocity than stroke-operated male mice, who, in comparison, swam longer distances with higher velocity than sham-operated male mice. As other parameters were similar between groups, this difference likely relates to locomotor behavior rather than spatial memory.

Home-cage locomotion was monitored 24/7 in digital ventilated cages (DVCs). During day and night, APP/PS1 mice displayed increased locomotion compared to WT mice, while stroke-induced animals exhibited greater locomotion than their sham counterparts. Interestingly, female APP/PS1 mice displayed hyperactive behavior, walking nearly twice the distance of their male and WT counterparts, suggesting increased energy expenditure [50]. This hyperactive phenotype aligns with clinical observations where female AD patients exhibit more severe motor behaviors and a tendency to pace and wander [51,52]. Importantly, the elevated activity in stroke-operated mice indicates stroke-related neurotransmitter imbalances and increased locomotor behavior [53,54,55].

### 4.2. Body Weight

Body weight loss is common in both IS and AD [56,57]. In this study, despite initial recovery, stroke-operated WT mice showed lower body weight compared to sham-operated mice from 6 months post-stroke. This stroke effect was absent in APP/PS1 mice, likely due to less constant weight gain, typical for this AD-like animal model [58].

Noteworthy, female APP/PS1 mice were lighter than female WT mice, whereas male APP/PS1 mice were heavier than their WT counterparts. In contrast, Pugh et al. reported weight loss in both sexes at 10 months of age [58]. This sex-specific weight difference in APP/PS1 mice may reflect more pronounced hyperactivity in female compared to male APP/PS1 animals, as indicated by higher locomotor activity in their home cages (DVCs). In AD patients, hyperactivity has been linked to increased energy expenditure and weight loss despite sufficient or excessive caloric intake [56,59]. Furthermore, a negative correlation between body mass index and abnormal motor behavior has been observed in AD [56,60]. Nevertheless, while male APP/PS1 mice also showed increased locomotion compared to WT males, their body weight was significantly higher. Given the conflicting data on body weight changes in APP/PS1 mice, further studies are needed to clarify the interplay between locomotion, metabolic regulation, food intake, and body weight regulation in AD and stroke.

### 4.3. Cerebral Blood Flow

Midlife hypertension is a significant modifiable risk factor for both AD and IS [61]. We found elevated systolic blood pressure in APP/PS1 mice, notably at earlier ages than in other preclinical studies [62]. A recent study has shown that soluble Aβ triggers hypertension by causing endothelial dysfunction and vascular hypercontractility, leading to enhanced vasoconstriction and increased arterial stiffness [63,64]. Subsequently, subtle impairments in cerebrovascular autoregulation and reduced cerebral blood flow (CBF) were detectable in APP/PS1 mice even before the appearance of plaque deposits [62,65].

Patients with AD show CBF reductions years before cognitive decline and correlate with symptom severity [7,66]. In our study, we evaluated CBF at different stages of AD pathology, corresponding to subclinical (4 months of age, 2 months post-stroke), mid-clinical (8 months of age, 4 months post-stroke), and late-clinical (12 months of age, 8 months post-stroke) stages in humans. APP/PS1 mice exhibited decreased CBF in the contralateral thalamus of sham-operated mice and sporadically in the ipsilateral thalamus and cortex post-surgery. Surgery-related interhemispheric CBF differences may have masked genotype variations.

(Pre-)clinical studies have shown that stroke causes acute hypoperfusion in the infarct core, leading to neuronal death, and affects not only the ipsilateral hemisphere but also remote regions in the contralateral hemisphere [67]. While rodent studies of post-stroke hemodynamics have focused on the acute period (<1 month post-stroke), longitudinal assessments (>1 month post-stroke) are scarce. In this study, we found constant lower CBF post-surgery in the ipsilateral cortex and hippocampus. With increasing age and AD progression, contralateral hippocampal CBF decreased, while CBF in all other regions remained stable. Depending on the severity of the stroke, regional hypoperfusion can persist for weeks, but recovery is possible due to collateral blood supply [68]. We observed thalamic hypoperfusion 2 weeks post-stroke, followed by hyperperfusion at 4 and 8 months in the thalamus of stroke-operated mice. Similarly, hyperperfusion was measured in the hippocampus of stroke-induced mice 4 months post-surgery. A preclinical stroke study in rats reported that thalamic hyperperfusion correlated with increased blood vessel density and angiogenesis, which helps remove necrotic tissue and supports brain repair [65]. Ischemia-induced neurovascular coupling increases CBF to meet oxygen demands by dilating capillaries, creating hyperemia as a compensatory mechanism [69]. Considering the relatively minor stroke-related longitudinal changes in CBF in regions that play a distinctive role in spatial memory, it may be an explanation for why stroke-operated mice did not display deficits in spatial learning and memory. Furthermore, human and animal studies indicate that cerebral hypoperfusion acerbate Aβ pathology and vice versa, yet we did not find an interaction between genotype and surgery regarding CBF [70].

Regarding sex differences, premenopausal women exhibit higher CBF than postmenopausal women and age-matched men. Estrogen can protect against stroke as it promotes cerebral perfusion by decreasing vasoconstriction [20]. Our findings contrast with previous studies, showing that female mice, independent of the genotype, consistently display lower CBF in several regions, both early after stroke and at a young age (cortex), as well as late post-stroke and with advancing age (cortex, hippocampus, and thalamus). Aβ deposition begins earlier in female APP/PS1 mice, which also show a higher parenchymal and vascular Aβ burden, impairing vascular function and leading to brain hypoperfusion, as supported by Waigi et al., who found that Aβ plaques colocalized with endothelial cells in the hippocampus of female, but not male, APP/PS1 mice [64,65,71]. Amyloid overexpression is known to disrupt endothelial function via reactive oxygen species (ROS), altered endothelial nitric oxide synthase (eNOS) activity, and impaired vascular regulation resulting from endothelial dysfunction rather than structural damage or apoptosis [64,72]. Accordingly, we found higher Aβ burden in the hippocampus of female mice compared to male mice at 12 months of age, as well as reduced CBF, particularly in female APP/PS1 compared to their male littermates in the ipsilateral thalamus (0.5 months post-stroke) and in the hippocampus (8 months post-stroke). In conclusion, female mice may be more susceptible to Aβ-related vascular dysfunction due to higher amyloid burden at an earlier age.

### 4.4. Structural Changes

Spatial and contextual learning have been shown to be dependent on corticohippocampal activity [73]. While hippocampal volume and cortical thickness decrease with age, both regions are particularly vulnerable to accelerated atrophy from hypoperfusion and AD pathology, leading to cognitive decline that exceeds typical age-related impairments [74]. IS often causes secondary neurodegeneration in the hippocampus due to disconnection from the infarcted site, as evidenced in rodent studies [75,76]. Accordingly, in this study, stroke-operated mice had lower ipsilateral hippocampal volume, which further decreased over time, as well as decreased cortical thickness in the ipsilateral hemisphere compared to sham-operated mice. Nevertheless, despite these structural damages, the learning ability remained unaffected, suggesting brain resilience to ischemia or the ability of functional recovery.

Previous findings indicate that both AD patients [77] and APP/PS1 mice [78] exhibit cortical and hippocampal atrophy. In contrast, we found increased cortical thickness in APP/PS1 mice at 4 and 8 months post-surgery. Several mechanisms could explain this seemingly counterintuitive result. Aβ accumulation may trigger neuroinflammation, causing local fluid increases or glial recruitment, leading to an overestimation of cortical thickness [79,80]. Additionally, Aβ plaques could artificially inflate cortical thickness due to their space-occupying nature. Supporting this, preclinical studies demonstrated that cerebral hypoperfusion can accelerate regional Aβ concentrations, which may contribute to increased cortical thickness [81]. Additionally, clinical trials show cortical thinning following Aβ-lowering therapies [82].

Furthermore, cortical thickness was increased in female mice, consistent with clinical studies [83]. Sex-based disparities in cortical thickness have rarely been studied in rodents, and data on APP/PS1 mice are limited. However, a few preclinical studies suggest thicker cortices in males compared to females [84,85]. The increased cortical thickness in female mice may result from amyloid accumulation starting at a younger age compared to male mice [86]. Female APP/PS1 mice also exhibited greater amyloid deposition than males, which could artificially increase cortical thickness as described previously [81,87]. Likewise, hippocampal volumes were also larger in female APP/PS1 mice compared to WT mice, indicating the deleterious effect of Aβ pathology in female animals.

### 4.5. White Matter Integrity

Both vascular changes and Aβ-dependent neuronal dysfunction result in loss of structural and functional connectivity [88]. Likewise, IS not only causes GM damage defined by neuronal necrosis but also elicits WM injury, leading to cognitive impairment. Therefore, we assessed regional microstructural integrity using longitudinal in vivo DTI and post-mortem PLI. Fractional anisotropy (FA) is a sensitive measure of myelination, axonal degeneration, and cytoskeletal features [89]. PLI retardance values are representative of myelination, with decreased values suggesting myelin loss [42].

Early postoperative FA values (0.5 to 4 months post-stroke) showed increases in FA in the GM, corpus callosum (CC), fornix, and ipsilateral hippocampus, indicative of WM recovery. In the early stages of AD or following IS, compensatory mechanisms, such as axonal sprouting or remyelination, may help maintain or even temporarily increase FA values as the brain attempts to reorganize and preserve its structural integrity [90]. However, with advanced age and AD pathology, between 4 and 8 months post-surgery, FA decreased in the WM, GM, CC, fornix, and ipsilateral cortex across all experimental groups. Although hippocampal FA increased over time, we observed continuous interhemispheric differences in the hippocampus, indicating a marked impact of the surgeries causing microstructural abnormalities measured via innovative PLI. Several studies convergently report reduced FA in the chronic stage post-stroke, which has been linked to demyelination [91,92,93,94,95]. However, PLI results did not resemble surgery-related interhemispheric differences in the hippocampus, as neither myelin density nor fiber orientation were affected. Thus, myelin structure and organization appear to be unchanged in the hippocampus 12 months after stroke induction. This suggests that FA reductions in the hippocampus arise from other pathological mechanisms, such as axonal degeneration, swelling, or gliosis [96,97].

Furthermore, our findings revealed increased cortical FA in stroke-operated mice, paralleled by increasing FA in stroke-operated WT mice over time. A serial DTI study with experimental IS models showed an initial decrease in FA followed by normalization or elevation in perilesional WM, suggesting remyelination or axonal reorganization over time [92]. However, similar to our observations in the hippocampus, PLI did not show changes in myelin density or fiber orientation. This suggests that the observed FA increase is more likely attributable to the aforementioned pathological mechanisms.

As AD pathology advances with age, APP/PS1 mice demonstrated diminished FA values in the hippocampus and cortex compared to WT mice. These genotype differences, validated by post-mortem PLI, reveal decreased myelin density in the hippocampus and thalamus, aligning with previous DTI studies [33,98,99]. Noteworthy, stroke may accelerate Aβ accumulation and WM lesions by interfering with clearance pathways, explaining the very early WM degeneration in APP/PS1 mice [100].

Sex differences in WM integrity appear to be related to differences observed in CBF measurements. WM tracts are particularly vulnerable to the vascular damage induced by chronic hypoperfusion [101]. Additionally, higher Aβ plaque load in female APP/PS1 mice may contribute to these differences. Female mice exhibited compromised cortical WM, while males displayed cortical WM recovery over time. This corresponds with CBF measurements, where cortical CBF was notably lower in female mice compared to males. Post-mortem, female APP/PS1 mice displayed decreased cortical myelin density compared to males, consistent with findings from Zhou et al., who reported lower total WM volume and reduced myelinated fiber parameters in female APP/PS1 mice [102]. In addition, female APP/PS1 mice also showed lower myelin density than female WT mice, whereas the reverse was true for males. A new mouse model for AD with chronic hypoperfusion also identified hippocampal WM lesions from 6 months of age [103], further supporting the notion that female mice are more susceptible to AD-related WM changes due to increased hypoperfusion.

In contrast, female mice exhibited higher myelin density in the thalamus than males despite having lower CBF. However, this increased density was accompanied by greater thalamic dispersion in females, suggesting worse myelin quality. This finding may indicate that while remyelination may occur, fiber organization is less structured.

### 4.6. Gray Matter Integrity

To assess the impact of AD and IS on GM integrity, we used mean diffusivity (MD) measured via in vivo DTI. Post-stroke, increased MD indicates edema, cellularity, and necrosis in the GM [104]. Post-mortem PLI dispersion provides quantitative estimates of fiber dispersal, with increased values indicating worse myelin quality [40].

Long-term post-surgery, we observed a significant decrease in MD in the cortex, suggesting GM recovery. A decrease in MD in structures such as the corpus callosum (CC) and fornix, which are primarily WM, often indicates recovery processes such as remyelination, reduced edema, or axonal reorganization. Despite the improvement in GM integrity, surgery-related interhemispheric differences with increased MD in the ipsilateral hemisphere appeared early in the hippocampus and persisted in the cortex, indicating continuous GM deterioration.

Most stroke studies report an initial decrease in MD shortly after stroke due to edema [41,104], followed by a persistent increase in MD in the chronic stage [105]. In agreement, we found decreased MD in the hippocampus of stroke-operated mice 0.5 months after surgery. Additionally, sex and genotype affected hippocampal GM integrity shortly after surgery. Female APP/PS1 mice had higher MD than female WT, while female WT had lower MD than male WT, indicating better GM integrity in female WT mice, but AD-related deterioration in female APP/PS1 mice. The in vivo data suggest that sex, IS, and AD pathology affect GM integrity only shortly after stroke, with no long-term differences observed, indicating the brain’s capacity to recover.

PLI, offering higher spatial resolution than in vivo DTI, provides detailed insights into myelin quantity and fiber orientation, thereby enhancing the understanding of anatomical changes [40]. Therefore, although not detected in vivo, thalamic dispersion levels indicated stroke-related deterioration of myelin quality in the thalamus 8 months post-surgery. Unexpectedly, post-mortem PLI analysis showed that female APP/PS1 mice had lower cortical dispersion values compared to male APP/PS1 and female WT mice, indicating better myelin quality in female APP/PS1. However, the combination of decreased dispersion and myelin density in female APP/PS1 mice may suggest increased cellular density due to inflammatory responses, along with disruption of organized WM tracts. In the CC, female mice showed lower dispersion than males, indicating better myelin quality, while myelin density did not differ. The improved GM integrity in the CC may be maintained by the neuroprotective effects of estrogen [20].

### 4.7. Neuroinflammation

Acute brain inflammation may protect against ischemic injury and Aβ plaques, but the imbalance of pro- and anti-inflammatory signals in advanced AD and IS leads to chronic inflammation [106,107]. In APP/PS1 mice, microglial activation typically coincided with Aβ plaques development [108]. Microglia and astrocytes clear Aβ but release proinflammatory cytokines, which attract more microglia [109]. Notably, vascular Aβ deposition causes vascular inflammation and weakens the blood–brain barrier [110]. Consistent with previous studies, we found increased neuroinflammation in APP/PS1 mice, as assessed by IBA-1 immunoreactivity in the cortex and the hippocampus [62,111].

In IS, microglia rapidly activate to clear debris and release repair factors, facilitating initial recovery, while chronic inflammation can worsen pathology [106]. In our study, we observed no stroke-related increase in IBA-1+ cells but an increased IBA-1+ area in the hippocampus of stroke-operated male mice and in the thalamus of stroke-operated WT mice compared to their sham-operated littermates. The findings indicate that the count of microglia is not elevated; rather, existing microglia are activated, resulting in increased cell area due to soma enlargement [112]. This highlights the presence of chronic neuroinflammation even 8 months post-surgery, consistent with studies examining neuroinflammation up to 5 weeks post-stroke [30,31,93,113]. Sustained neuroinflammation can lead to further neuronal damage and accelerate AD pathology.

Importantly, we detected clear sex differences in neuroinflammation, with female mice showing less neuroinflammation in the hippocampus and cortex compared to males. Female stroke mice had a lower IBA-1+ area in the ipsilateral hippocampus, suggesting a neuroprotective advantage in females, possibly related to the role of estrogen in modulating microglial activation [114]. These findings suggest reduced neurodegenerative damage in females post-stroke and highlight the impact of hormonal differences on brain inflammation.

### 4.8. Limitations

A limitation of this study is that in some cases of stroke, complete degeneration of the ipsilateral hippocampus resulted in the exclusion of these animals from the analysis. Consequently, the true impact of stroke may be underestimated due to the unaccounted hippocampal degeneration.

Furthermore, there is a variation in sample size between experimental groups due to data loss or technical problems and, as mentioned before, complete degeneration of brain regions. This was particularly evident in the CBF measurements (Appendix A). This discrepancy may have reduced statistical power to detect subtle CBF differences and should be considered when interpreting the results. Future studies with more consistent data acquisition for all parameters will help to validate these findings.

## 5. Conclusions

This study highlights the bidirectional relationship between IS and AD and the significant impact of sex on these conditions. APP/PS1 mice suffered from severe cognitive impairment, hyperactivity, elevated blood pressure, reduced cerebral blood flow (CBF), and chronic neuroinflammation. Stroke exacerbated structural brain damage and induced chronic neuroinflammation, yet it did not cause cognitive impairment, possibly due to compensatory mechanisms. Female mice, although they exhibited hypoperfusion, higher Aβ levels, and greater structural changes, benefited from the neuroprotective effects of estrogen, resulting in less neuroinflammation.

Our results show the necessity of considering sex differences in AD and IS research to develop more effective therapeutic strategies. Although sex did not affect cognitive performance in this study, differences were present regarding CBF, neuroinflammation, and structural changes. Therefore, therapeutic approaches need to be tailored to differences in disease progression and pathology between men and women. Incorporating sex as a biological variable in preclinical and clinical research ensures a more complete understanding of these diseases, enhancing the development of targeted, effective treatments for diverse patient populations.

## Figures and Tables

**Figure 1 life-15-00333-f001:**
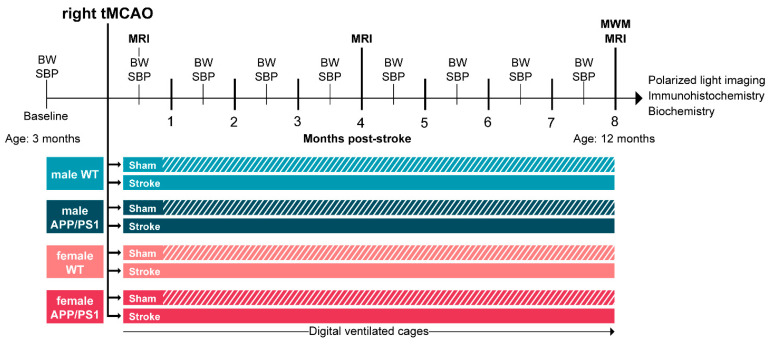
Overview of the study design. This longitudinal study investigated the impact of stroke and Alzheimer’s disease on cognitive impairment, emphasizing sex differences. Male and female WT and APP_swe_/PS1_dE9_ (APP/PS1) mice (3-month-old) were included. Baseline physiological parameters (body weight (BW) and systolic blood pressure (SBP)) were monitored. At 3.5 months of age, mice underwent either right transient middle cerebral artery occlusion (tMCAO) or sham surgery, resulting in eight groups: (1) male WT sham, (2) male WT stroke, (3) male APP/PS1 sham, (4) male APP/PS1 stroke, (5) female WT sham, (6) female WT stroke, (7) female APP/PS1 sham, and (8) female APP/PS1 stroke. Body weight and systolic blood pressure were monitored monthly post-stroke. Walking patterns of each mouse were individually monitored 24/7 for eight months using digital ventilated cages (DVCs). Magnetic resonance imaging (MRI) was conducted at 0.5, 4, and 8 months post-stroke. At 12 months of age (8 months post-stroke), spatial learning and memory were assessed using the Morris water maze (MWM) test. After the final neuroimaging session, mice were sacrificed, and brains were collected for post-mortem analysis, including immunohistochemical stainings, polarized light imaging, and biochemistry.

**Figure 2 life-15-00333-f002:**
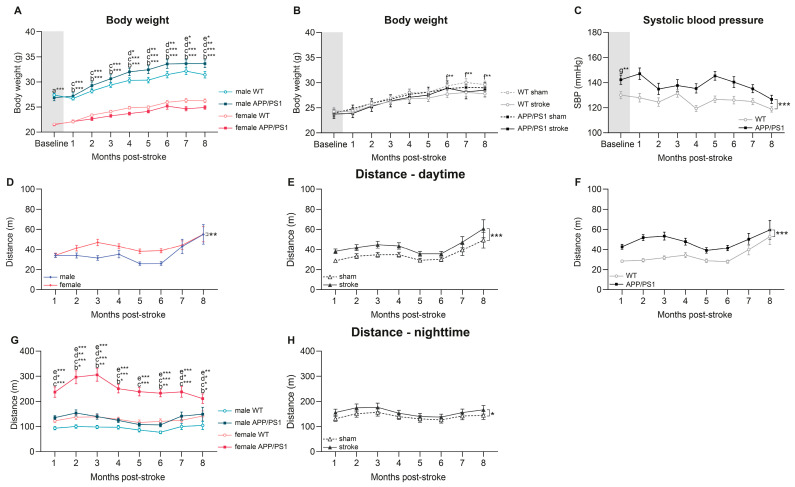
Monthly (**A**,**B**) body weight and (**C**) systolic blood pressure (SBP) measurements were performed from baseline over a period of 8 months, following either sham or stroke surgery in both male and female WT and APP/PS1 mice. (**A**) Prior to stroke (baseline), at 3.5 months of age, female mice had a lower body weight than male mice (a: male vs. female; male n = 42, female n = 49). Across the 8 months post-surgery, male WT and male APP/PS1 mice consistently gained weight. Female WT and APP/PS1 mice exhibited less weight gain than males, with female mice generally being lighter than males (b: female WT vs. male WT, c: male APP/PS1 vs. female APP/PS1). From month 4, male APP/PS1 mice outweighed male WT mice, while female APP/PS1 mice were lighter than female WT mice, especially in the last two months (d: male APP/PS1 vs. male WT, e: female APP/PS1 vs. female WT; male WT n = 27, male APP/PS1 n = 15, female WT n = 29–30, female APP/PS1 n = 19). (**B**) From 6 months post-stroke onward, WT stroke mice displayed lower body weight compared to the control group that underwent sham surgery (f: WT stroke vs. WT sham; WT sham n = 29, WT stroke n = 27–28, APP/PS1 sham n = 17, APP/PS1 stroke n = 17). (**C**) At baseline, APP/PS1 mice had higher systolic blood pressure (SBP) than WT mice (g: APP/PS1 vs. WT). Post-surgery, APP/PS1 mice continued to exhibit higher SBP compared to their WT counterparts (WT n = 42–51, APP/PS1 n = 25–34). Furthermore, we monitored the distance the animals walked in their home cages during both day and night. During daytime, we observed that (**D**) female mice walked more than males (male n = 27–32, female n = 32–42), (**E**) stroke-operated mice walked more than sham-operated mice (sham n = 29–33, stroke n = 30–41), and (**F**) APP/PS1 mice walked more than WT mice (WT n = 39–45, APP/PS1 n = 20–29). (**G**) During nighttime, female WT mice walked more than male WT mice (b: female WT vs. male WT), female APP/PS1 mice walked more than male APP/PS1 mice (c: male APP/PS1 vs. female APP/PS1), male APP/PS1 walked more than male WT mice (d: male APP/PS1 vs. male WT), and female APP/PS1 walked more than female WT (e: female APP/PS1 vs. female WT; male WT n = 23–24, male APP/PS1 n = 10–14, female WT n = 22–27, female APP/PS1 n = 13–18). (**H**) Moreover, stroke-operated mice consistently walked longer distances than sham-operated mice (sham n = 29–33, stroke n = 30–41). Data are presented as mean ± SEM. Significance is denoted as * *p* < 0.05, ** *p* < 0.01, *** *p* < 0.001.

**Figure 3 life-15-00333-f003:**
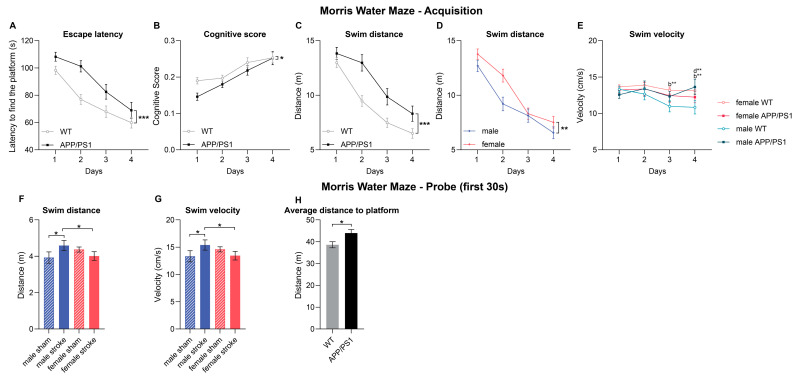
Assessment of spatial learning and memory in the Morris water maze (MWM). This figure illustrates the results of the Morris water maze test performed on male and female, sham- or stroke-induced APP/PS1 and WT mice, 8 months post-surgery at 12 months of age. Results from the 4-day acquisition phase, assessing spatial learning abilities, are depicted in the upper panel: (**A**) The escape latency, a measure of spatial learning ability, decreased in WT and APP/PS1 mice. However, APP/PS1 mice exhibited longer times to find the hidden platform compared to WT mice, a finding indicative of AD-related decline in spatial learning (WT n = 55, APP/PS1 n = 25). (**B**) Cognitive scores increased but were lower in APP/PS1 mice compared to WT mice (WT n = 55, APP/PS1 n = 25). (**C**) Moreover, APP/PS1 mice swam longer distances than WT mice (WT n = 55, APP/PS1 n = 25). (**D**) In terms of sex differences, female mice swam further distances than male mice (male n = 38, female n = 42). (**E**) Male WT mice exhibited a decrease in swim velocity over time, while female WT mice swam faster than their male counterparts on days 3 and 4. On day 4, male APP/PS1 mice swam faster than WT males (male WT n = 26, male APP/PS1 n = 12, female WT n = 29, female APP/PS1 n = 13) (b: female WT vs. male WT, d: male APP/PS1 vs. male WT). Results of the probe trial during its first 30 s, assessing spatial memory abilities, are shown in the lower panel. Female mice that underwent stroke surgery (**F**) swam shorter distances and (**G**) swam slower than their male counterparts. Conversely, male stroke mice exhibited an increased (**H**) swim distance and (**F**) swim velocity compared to male sham-operated mice (male sham n = 16, male stroke n = 16, female sham n = 20, female stroke n = 17). (**H**) Additionally, APP/PS1 mice maintained a greater distance from the former platform location than WT mice, suggesting a diminished spatial memory associated with AD (WT n = 46, APP/PS1 n = 23). Data are presented as mean ± SEM. Significance is denoted as * *p* < 0.05, ** *p* < 0.01, *** *p* < 0.001.

**Figure 4 life-15-00333-f004:**
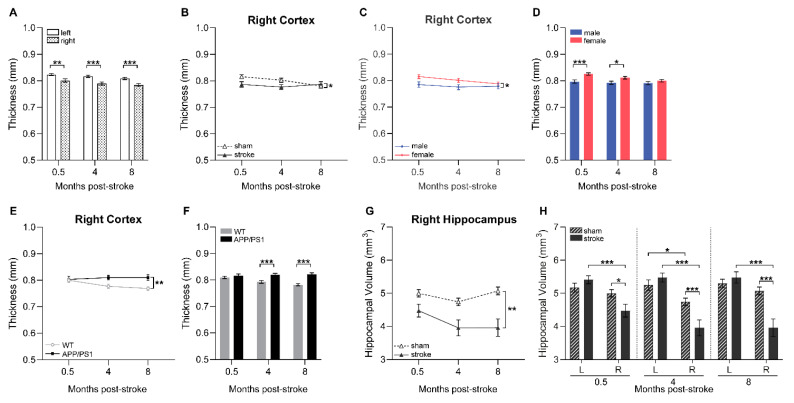
Longitudinal analysis of cortical thickness and hippocampal volume in male and female WT and APP/PS1 mice assessed at 0.5, 4, and 8 months post-stroke or -sham surgery. (**A**) Cortical thickness in the right (surgery-affected) hemisphere was consistently smaller than in the left hemisphere across all timepoints (right n = 86–90, left n = 86–90). (**B**) Overall, stroke-operated animals displayed a thinner right cortex compared to sham-operated animals (sham n = 44–45, stroke n = 42–45). (**C**) Over the 8-month post-surgery period, the right hemisphere’s cortex was consistently thicker in female compared to male mice (male n = 39–42, female n = 47–49). (**D**) Female mice exhibited a thicker cortex than male mice at 0.5 and 4 months post-surgery (male n = 78–84, female n = 94–98). (**E**) APP/PS1 mice had a thicker right cortex compared to WT mice (WT n = 55–56, APP/PS1 n = 31–34). (**F**) This effect was particularly pronounced at 4 and 8 months post-stroke, where APP/PS1 mice displayed greater cortical thickness than WT mice (WT n = 110–112, APP/PS1 n = 62–68). (**G**) Regarding hippocampal volume, stroke-operated mice exhibited lower right hippocampal volume compared to sham-operated mice (sham n = 45–46, stroke n = 43–45). (**H**) This stroke-induced reduction in hippocampal volume was also observed when considering the combined hippocampal volume at 0.5, 4, and 8 months post-surgery. Furthermore, the right hippocampal volume remained consistently smaller than the left hippocampal volume in stroke-operated mice. At 4 months after surgery, a similar interhemispheric difference was also measured among sham mice. (left sham n = 45–46, left stroke n = 43–45, right sham n = 45–46, right stroke n = 43–45). Data are presented as mean ± SEM. Significance is denoted as * *p* < 0.05, ** *p* < 0.01, *** *p* < 0.001.

**Figure 5 life-15-00333-f005:**
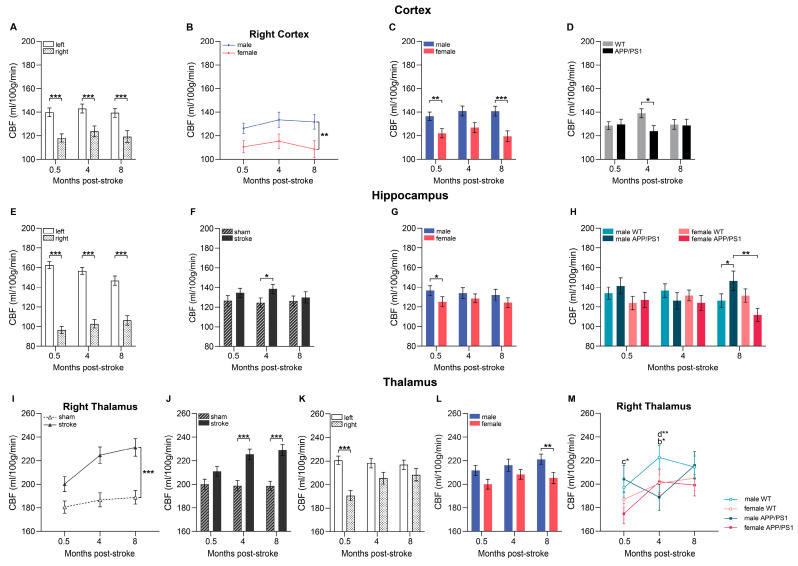
Longitudinal analysis of cerebral blood flow in male and female WT and APP/PS1 mice assessed at 0.5, 4, and 8 months post-stroke or -sham surgery. (**A**) Cortical CBF was consistently lower in the right hemisphere compared to the left at all time points (left n = 65–91, right n = 65–91). (**B**) Female mice had lower CBF in the right cortex compared to male animals during the post-surgery period (male n = 30–42, female n = 35–49). (**C**) At 0.5 and 8 months after stroke, female mice exhibited lower cortical CBF than males (male n = 60–84, female n = 70–98). (**D**) Four months after surgery, APP/PS1 mice had lower cortical CBF compared to their WT littermates (WT n = 90–114, APP/PS1 n = 40–68). (**E**) CBF in the hippocampus was significantly lower in the right hemisphere compared to the left at all time points (left n = 65–91, right n = 57–82). (**F**) At 4 months after surgery, stroke mice had higher CBF in the hippocampus than sham-operated animals (sham n = 70–92, stroke n = 52–86). (**G**) At 0.5 months after stroke induction, female animals showed lower CBF compared to males (male n = 56–81, female n = 66–93). (**H**) Male APP/PS1 mice had higher hippocampal CBF than male WT mice, whereas female APP/PS1 mice displayed lower CBF compared to male APP/PS1 mice (male WT n = 40–51, male APP/PS1 n = 16–30, female WT n = 42–56, female APP/PS1 n = 24–37). (**I**) In the right thalamus, stroke-operated mice displayed higher CBF over time compared to sham mice (sham n = 35–46, stroke n = 30–45). (**J**) Particularly at 4 and 8 months post-surgery, stroke-operated mice had higher thalamic CBF compared to sham-operated animals (sham n = 70–92, stroke n = 60–90). (**K**) Right thalamic CBF was lower than in the left hemisphere 0.5 months post-operation (left n = 65–91, right n = 65–91). (**L**) At 8 months after surgery, female mice showed lower CBF than male mice (male n = 65–91, female n = 65–91). (**M**) At 0.5 months post-stroke, female APP/PS1 mice showed lower CBF in the right thalamus than male APP/PS1 mice. At 4 months post-surgery, male APP/PS1 mice displayed lower CBF than male WT mice, and among APP/PS1 animals, females exhibited lower CBF than males. Additionally, an increase in CBF was observed in the right thalamus of male WT and female APP/PS1 mice between 0.5 and 4 months after surgery (male WT n = 22–27, male APP/PS1 n = 8–15, female WT n = 23–30, female APP/PS1 n = 12–19) (b: female WT vs. male WT, c: female APP/PS1 vs. male APP/PS1, d: male APP/PS1 vs. male WT). Data are presented as mean ± SEM. Significance is denoted as * *p* < 0.05, ** *p* < 0.01, *** *p* < 0.001.

**Figure 6 life-15-00333-f006:**
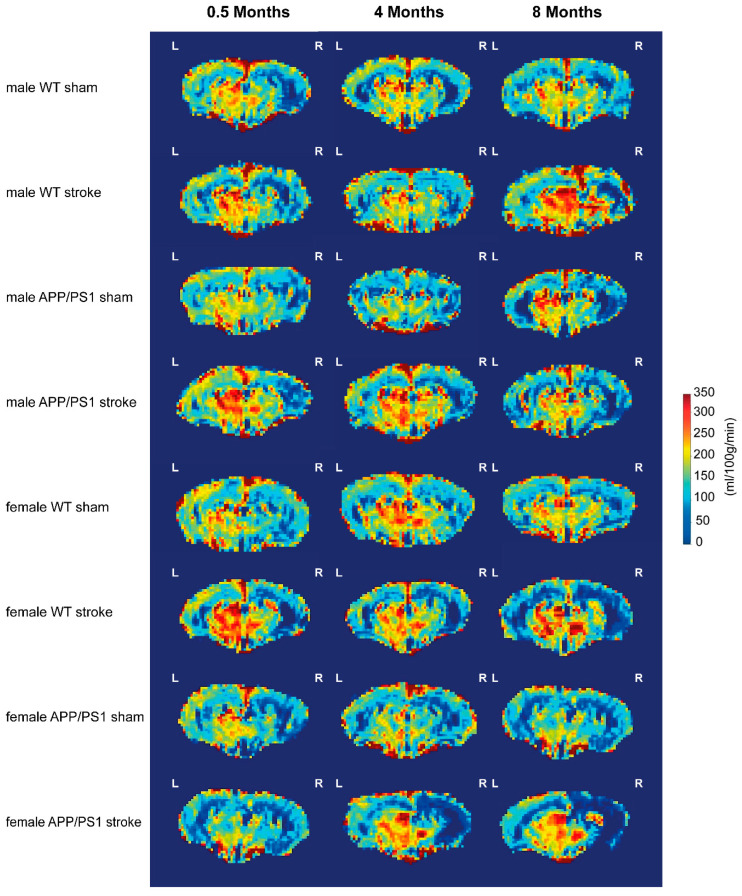
High-resolution voxel-wise CBF images at bregma level −1.94 in male and female WT and APP/PS1 mice, assessed at 0.5, 4, and 8 months post-stroke or -sham surgery.

**Figure 7 life-15-00333-f007:**
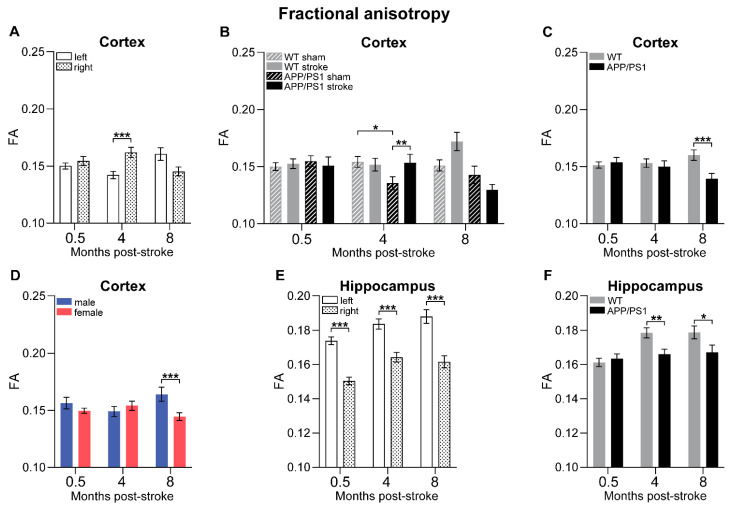
Longitudinal diffusion tensor imaging analysis. A comprehensive representation of fractional anisotropy (FA) and mean diffusivity (MD) in male and female WT and APP/PS1 mice assessed at 0.5, 4, and 8 months post-stroke or -sham surgery. (**A**) At 4 months post-stroke, the right cortex exhibited higher FA than the left cortex (left n = 67–76, right n = 67–76). (**B**) Comparing cortical FA at 4 months post-surgery, APP/PS1 sham mice showed lower FA than WT sham mice. In contrast, APP/PS1 stroke mice had higher cortical FA than APP/PS1 sham mice (WT sham n = 46–54, WT stroke n = 38–42, APP/PS1 sham n = 24–32, APP/PS1 stroke n = 22–32). (**C**) At 8 months post-stroke, APP/PS1 mice demonstrated lower cortical FA compared to WT mice (WT n = 88–94, APP/PS1 n = 46–64). (**D**) Additionally, at 8 months after surgery, female mice exhibited lower cortical FA than male mice (male n = 58–62, female n = 76–92). (**E**) Consistently, the right hippocampus showed lower FA than the left at all timepoints (left n = 67–76, right n = 67–76). (**F**) At both 4 and 8 months after stroke, the hippocampus of APP/PS1 mice displayed lower FA compared to that of WT mice (WT n = 88–94, APP/PS1 n = 46–64). (**G**) Across all time points, the right cortex consistently exhibited lower MD than the left cortex (left n = 67–76, right n = 67–76). (**H**) At 0.5 months post-stroke, the right hippocampus showed higher MD compared to the left hippocampus (left n = 67–76, right n = 67–76). (**I**) At the same time point, stroke mice displayed lower hippocampal MD than sham mice (sham n = 37–40, stroke n = 30–37). (**J**) Also, at 0.5 months after stroke, female WT mice demonstrated lower MD compared to male WT mice. Moreover, female APP/PS1 mice exhibited higher MD than their female WT counterparts (male WT n = 17–20, male APP/PS1 n = 10–13, female WT n = 25–27, female APP/PS1 n = 13–19). Data are presented as mean ± SEM. Significance is denoted as * *p* < 0.05, ** *p* < 0.01, *** *p* < 0.001.

**Figure 8 life-15-00333-f008:**
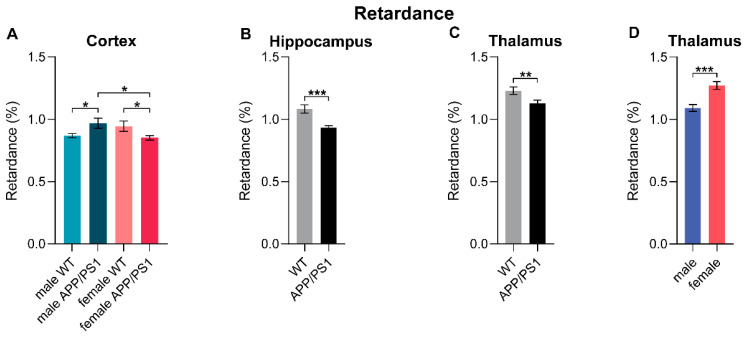
Polarized light imaging (PLI) was conducted on post-mortem brains to evaluate myelin density and fiber orientation across different regions of interest, including the cortex, corpus callosum, hippocampus, and thalamus. This analysis utilized retardance maps for myelin density, as shown in the upper panel, and dispersion maps for fiber orientation, as shown in the lower panel. PLI was performed on male and female WT and APP/PS1 mice 8 months after sham and stroke surgeries when the mice were 12 months old. Retardance serves as an indicator of myelination, with lower values suggesting potential myelin degradation. Similarly, dispersion levels are a quantitative assessment of fiber orientation, where lower dispersion indicates better myelin quality. (**A**) In the cortex, notable genotype differences were observed in both male and female animals. Myelin density was higher in male APP/PS1 mice compared to male WT mice, whereas female APP/PS1 mice had lower myelin density compared to female WT mice. Moreover, female APP/PS1 mice had lower myelin density than their male counterparts (male WT n = 40, male APP/PS1 n = 26, female WT n = 44, female APP/PS1 n = 28). (**B**) In the hippocampus, myelin density was lower in APP/PS1 mice compared to WT mice (WT n = 83, APP/PS1 n = 56). (**C**) The same genotype difference was also present in the thalamus, where APP/PS1 mice displayed lower myelin density compared to WT mice (WT n = 92, APP/PS1 n = 58). (**D**) Additionally, in the thalamus, female mice had higher myelin density than male mice (male n = 68, female n = 82). (**F**) In terms of dispersion values measured in the cortex, female APP/PS1 mice exhibited lower values compared to both female WT mice and male APP/PS1 mice (male WT n = 40, male APP/PS1 n = 26, female WT n = 44, female APP/PS1 n = 28). (**G**) In the corpus callosum, dispersion values were generally lower among female mice than male mice (male n = 70, female n = 80). (**H**) Conversely, in the thalamus, female mice showed higher dispersion levels compared to their male counterparts (male n = 68, female n = 82). (**I**) In the thalamus, dispersion values were higher among stroke-operated mice in comparison to their sham-operated littermates (sham n = 80, stroke n = 70). Representative images of (**E**) retardance and (**J**) dispersion maps. Data are presented as mean ± SEM. Significance is denoted as * *p* < 0.05, ** *p* < 0.01, *** *p* < 0.001.

**Figure 9 life-15-00333-f009:**
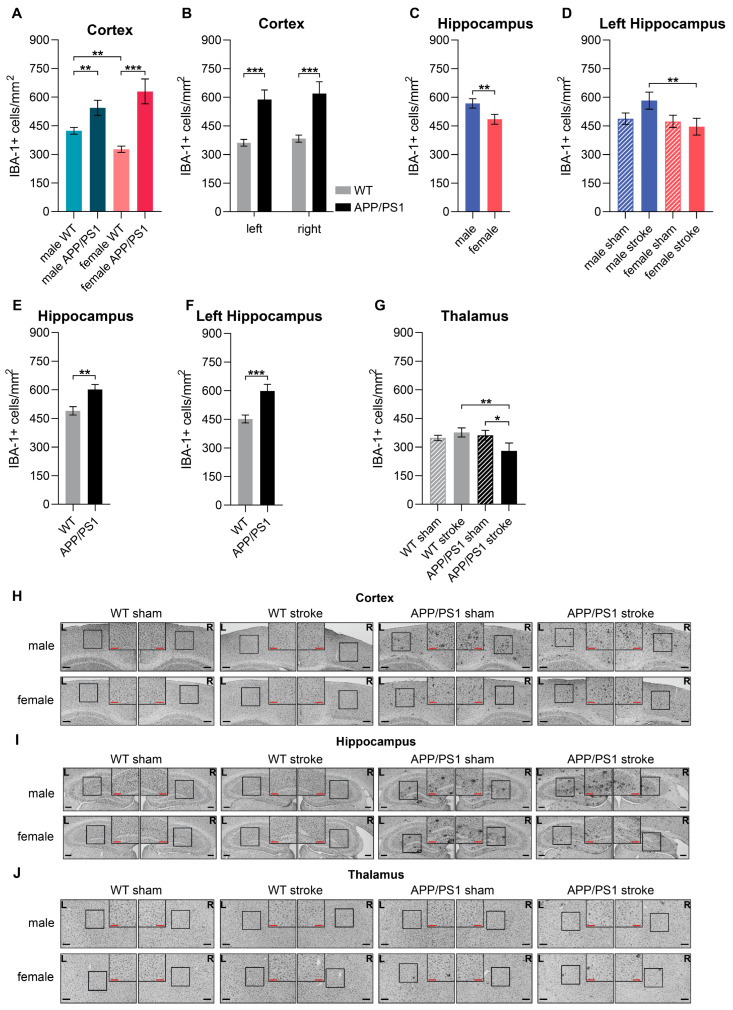
Immunohistochemical analysis of ionized calcium-binding adapter molecule 1 (IBA-1) as a measure of neuroinflammation. IBA-1+ microglia/macrophages were quantified across the cortex, hippocampus, and thalamus in male and female, sham- or stroke-operated WT and APP/PS1 mice, 8 months post-surgery at the age of 12 months. (**A**) Female WT mice displayed a lower amount of activated microglia in the cortex compared to male WT mice. Moreover, both female and male APP/PS1 mice exhibited a higher count of activated microglia in the cortex than their respective WT littermates (male WT n = 50, male APP/PS1 n = 22, female WT n = 56, female APP/PS1 n = 24). (**B**) In both cortical hemispheres, APP/PS1 mice had an increased amount of activated microglia compared to WT animals (left WT n = 53, left APP/PS1 n = 23, right WT n = 53, right APP/PS1 n = 23). (**C**) In the hippocampus, female mice displayed fewer activated microglia than male mice (male n = 73, female n = 80). (**D**) Notably, in the left hippocampus, female stroke mice displayed a lower amount of activated microglia compared to male stroke mice (male sham n = 18, male stroke n = 19, female sham n = 22, female stroke n = 18). (**E**) APP/PS1 mice also had a higher count of activated microglia in the combined hippocampus than WT mice (WT n = 105, APP/PS1 n = 48), (**F**) with this genotype effect being particularly detected in the left hippocampus (WT n = 53, APP/PS1 n = 24). (**G**) In the thalamus, stroke-operated APP/PS1 mice displayed a significantly lower count of activated microglia compared to both stroke-operated WT mice and sham-operated APP/PS1 mice (WT sham n = 52, WT stroke n = 52, APP/PS1 sham n = 16, APP/PS1 stroke n = 24). Representative images of IBA-1 staining in the left (L) and right (R) (**H**) cortex, (**I**) hippocampus, and (**J**) thalamus The boxes indicate regions that are visualized at higher magnification in the upper corners. (black scale bar = 200 µm; red scale bar: 100 µm). Data are presented as mean ± SEM. Significance is denoted as * *p* < 0.05, ** *p* < 0.01, *** *p* < 0.001.

**Figure 10 life-15-00333-f010:**
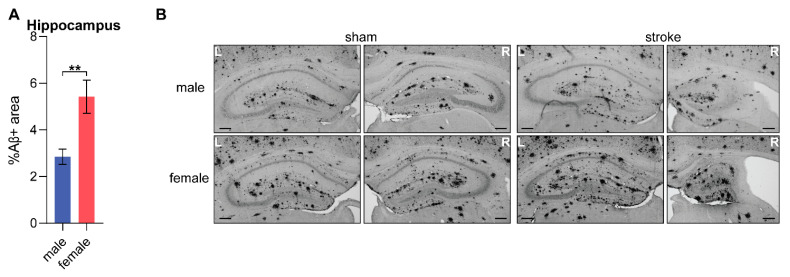
(**A**) Immunohistochemical analysis of amyloid beta (Aβ) in the hippocampus of male and female, sham- or stroke-operated APP/PS1 mice, 8 months post-surgery at the age of 12 months. The relative area covered with Aβ deposition was larger in female mice compared to male mice (male n = 30, female n = 36). (**B**) Representative images of WO-2 staining in the hippocampus (black scale bar = 200 µm). Data are presented as mean ± SEM. Significance is denoted as ** *p* < 0.001.

## Data Availability

The original data presented in the study are openly available on FigShare at https://doi.org/10.6084/m9.figshare.28387979.v1 (accessed on 20 February 2025).

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
