# Peer review of "Sex-Specific Adaptations in Alzheimer’s Disease and Ischemic Stroke: A Longitudinal Study in Male and Female APPswe/PS1dE9 Mice"

_life, 2025, doi:10.3390/life15030333_

Round 1
Reviewer 1 Report
Comments and Suggestions for Authors
The study examines the long-term impact of stroke on Alzheimer's disease progression AD mice model and found that APP/PS1 mice exhibited cognitive decline, white matter degeneration, and decreased myelin density, while stroke-operated mice retained cognition. Female APP/PS1 mice had more severe Aβ depositions, hyperactivity, lower body weight, and reduced cerebral blood flow, suggesting potential neuroprotection. Sex-specific therapies are crucial for both AD and stroke.
Recommendation
- The article is well experimentally planned, and organized, and has a novel impact.
- The introduction is sufficient and the methods are written widely with proper citations.
- Results texts are corroborated with respective figures and written well to make the reader understand.
- Supplementary data is sufficient and helpful in providing readers with a clearer understanding.
- The discussion part is nicely written however it needs some new citations to support the data in some places.
Minor suggestions
- Lines 891-908, it would be more effective to discuss these results in comparison with published articles to enhance the analysis.
- In the discussion section, some cited papers are nearly 20 years old (for example, citations 83, 95, and 104.). It would be more effective to compare your outcomes with recently published articles.
- lines 1084-1091, if any published articles are presenting the same results, it would be beneficial to corroborate the findings with them to strengthen the argument.
- Check the reference pattern and correct it accordingly.
Author Response
Please see the attachment.
For readability:
In the point-by-point response, the reviewer's comments are in bold black font, followed by our answers in plain font. The corresponding changes added to the manuscript are shown in blue italic font. Adjustments within the manuscript itself are added with tracked changes.
Reviewer 1:
We thank the reviewer for their clear summary and appreciation of our study’s novelty and organization. In response to your comments, we have made the following revisions:
- Lines 891-908, it would be more effective to discuss these results in comparison with published articles to enhance the analysis.
Reply: We thank the reviewer for this suggestion. The section between lines 891-908 is intended to provide a concise summary of the main results, while a detailed discussion, including comparisons with published articles, follows in the following discussion section. We agree that without explicitly stating its purpose, this section may have been misleading for the reader. To enhance clarity and improve readability, we have now revised and shortened the text. We hope this adjustment better aligns with the reviewer’s expectations.
(p.27 lines 917-931) This study identified several important findings. APP/PS1 mice exhibited impairments in spatial learning and memory, hyperactive behavior, hypertension, white matter (WM) degeneration, increased cortical thickness, and chronic neuroinflammation.
Stroke reduced hippocampal volume and cortical thickness in the ipsilateral hemisphere, as well as temporary regional hyperperfusion. Both sham and stroke surgeries caused longitudinal hypoperfusion and diminished GM integrity. Stroke in particular caused cortical thinning, hippocampal atrophy, and increased mean. Post-mortem, stroke mice had reduced myelin fiber quality without evident myelin loss. WM initially recovered but degenerated in all groups long-term post-surgery. Despite marked structural deterioration, cognition remained intact, however, stroke mice were.
Female mice exhibited lower CBF and higher Aβ burden , implicating greater susceptibility to Aβ-related vascular dysfunction. Female APP/PS1 mice showed increased cortical thickness and hippocampal volumes, possibly due to early Aβ deposition. They also showed more hyperactivity, reflecting severe motor symptoms in female AD patients, but less neuroinflammation, possibly due to neuroprotective effects of estrogen.
- In the discussion section, some cited papers are nearly 20 years old (for example, citations 83, 95, and 104.). It would be more effective to compare your outcomes with recently published articles.
Reply: We agree with the reviewer’s suggestion and have updated several references with more recent papers to ensure our discussion reflects the latest research. However, some older references were retained because they provide fundamental or scarce findings, particularly in areas such as longitudinal MRI studies in mice. For instance, we have kept reference (ref. 85) by van der Zijden et al. (2008) in the paper, as it remains a foundational and highly relevant publication in this field. Despite its age, it provides essential insights that have not been extensively replicated or replaced by more recent studies - an area where more recent studies remain limited. Given the lack of comparable research, this reference continues to be a valuable contribution to our discussion.
Removed from the manuscript (83): Plaschke, K., et al., The effect of stepwise cerebral hypoperfusion on energy metabolism and amyloid precursor protein (APP) in cerebral cortex and hippocampus in the adult rat. Annals of the New York Academy of Sciences, 1997. 826: p. 502-506. replaced by: Yang, H., et al., The effect of chronic cerebral hypoperfusion on amyloid-β metabolism in a transgenic mouse model of Alzheimer’s disease (PS1V97L). Journal of Alzheimer's Disease, 2018. 62(4): p. 1609-1621.
Kept in the manuscript (95): van der Zijden, J.P., et al., Longitudinal in vivo MRI of alterations in perilesional tissue after transient ischemic stroke in rats. Experimental neurology, 2008. 212(1): p. 207-212.
Remove from manuscript (104): Black, S., F. Gao, and J. Bilbao, Understanding white matter disease: imaging-pathological correlations in vascular cognitive impairment. Stroke, 2009. 40(3_suppl_1): p. S48-S52. replaced by: Chen, X., et al., White matter damage as a consequence of vascular dysfunction in a spontaneous mouse model of chronic mild chronic hypoperfusion with eNOS deficiency. Molecular psychiatry, 2022. 27(11): p. 4754-4769.
- lines 1084-1091, if any published articles are presenting the same results, it would be beneficial to corroborate the findings with them to strengthen the argument.
Reply: We appreciate the reviewer’s suggestion to corroborate our findings with previously published studies. We agree that integrating relevant literature strengthens our argument and provides a broader context for our results. In response, we have revised the text to explicitly connect our findings with previous research.
Specifically, we have incorporated findings from Zhou et al. (ref. 102), who reported lower total WM volume and reduced myelinated fiber parameters in female APP/PS1 mice. This aligns with our observations of decreased cortical myelin density in female APP/PS1 mice compared to males, supporting the notion that sex differences in WM integrity are a feature of AD-related pathology.
Additionally, we included findings from a new AD mouse model with chronic hypoperfusion (ref. 103), which identified hippocampal WM lesions starting at 6 months of age. While this study does not directly examine sex differences, it reinforces the idea that hypoperfusion plays a critical role in WM vulnerability, particularly in the context of AD pathology. Given our results showing lower CBF and greater WM disruption in female APP/PS1 mice, this provides additional support for the hypothesis that females may be more susceptible to AD-related WM changes due to increased hypoperfusion.
(p. 30 lines 1105-1118) Sex differences in WM integrity appear to be related to differences observed in CBF measurements. WM tracts are particularly vulnerable to the vascular damage induced by chronic hypoperfusion [101]. Additionally, higher Aβ plaque load in female APP/PS1 mice may contribute to these differences. Female mice exhibited compromised cortical WM, while males displayed cortical WM recovery over time. This corresponds with CBF measurements, where cortical CBF was notably lower in female mice compared to males. Post-mortem, female APP/PS1 mice displayed decreased cortical myelin density com-pared to males, consistent with findings from Zhou et al., who reported lower total WM volume and reduced myelinated fiber parameters in female APP/PS1 mice [102]. In addition, female APP/PS1 mice also showed lower myelin density than female WT mice, whereas the reverse was true for males. A new mouse model for AD with chronic hypoperfusion also identified hippocampal WM lesions from 6 months of age [103], further supporting the notion that female mice are more susceptible to AD-related WM changes due to increased hypoperfusion.
- Check the reference pattern and correct it accordingly.
Reply: We apologize for the errors in the reference formatting. Some references were not properly embedded in EndNote, which led to incorrect citations. We have now updated the citation library, ensuring that all references are correctly formatted and accurately cited.

Reviewer 2 Report
Comments and Suggestions for Authors
The authors evaluated a spectrum of behavioral, molecular, and physiologic outcomes across sex in by inducing stroke in 3.5-month APPswe/PS1dE9 (APP/PS1) and wild-type mice. They examined the effect stroke on AD pathology on the following aspects: behavior, cerebral blood flow, structural integrity, myelin integrity, neuroinflammation and amyloid-beta deposition.
Overall, this is an important study that adds meaningfully to the available literature on sexual dimorphism in AD and stroke.This is a novel sex-based study on stroke and AD. The methods utilized were appropriately described.
CRITIQUE
1. For Morris water maze studies, the standard protocol includes 1-4 trial with inter trial interval of 1-5 min. Was there any specific reason to use 4 trial with inter trial interval of 1 hr ?
- For “cognitive scoring”, all the terms used for the search strategy should be described in short.
- Fig 2 B, Male and female should be labeled.
- Fig 2 B, C, E, F, H, label the specific time-points with * to represent the significant difference.
- Fig 6 images are blurred. If these can be replaced with a low zoomed images to make the fig clearer and more legible.
- How was no. of mice/group was decided? Power analysis?
